# Common virulence gene expression in adult first-time infected malaria patients and severe cases

J Stephan Wichers[1,2,3], Gerry Tonkin-Hill[4], Thorsten Thye[5], Ralf Krumkamp[5,6], Benno Kreuels[7,8,9], Jan Strauss[1,2,3†], Heidrun von Thien[1,2,3], Judith AM Scholz[1], Helle Smedegaard Hansson[10], Rasmus Weisel Jensen[10], Louise Turner[10], Freia-Raphaella Lorenz[11], Anna Schöllhorn[11], Iris Bruchhaus[1,3], Egbert Tannich[5,6], Rolf Fendel[11,12], Thomas D Otto[13], Thomas Lavstsen[10], Tim W Gilberger[1,2,3], Michael F Duffy[14], Anna Bachmann[1,2,3,6]*

[1]Molecular Biology and Immunology, Bernhard Nocht Institute for Tropical Medicine, Hamburg, Germany; [2]Centre for Structural Systems Biology, Hamburg, Germany; [3]Biology Department, University of Hamburg, Hamburg, Germany; [4]Wellcome Sanger Institute, Hinxton, United Kingdom; [5]Epidemiology and Diagnostics, Bernhard Nocht Institute for Tropical Medicine, Hamburg, Germany; [6]German Center for Infection Research (DZIF), Partner Site Hamburg-Lübeck-Borstel-Riems, Hamburg, Germany; [7]Department of Tropical Medicine, Bernhard Nocht Institute for Tropical Medicine, Germany , Hamburg, Germany; [8]Department of Medicine, College of Medicine, Blantyre, Malawi; [9]Department of Medicine, University Medical Center Hamburg-Eppendorf, Hamburg , Germany ; [10]CMP, University of Copenhagen, Copenhagen, Denmark; [11]Institute of Tropical Medicine, University of Tübingen, Tübingen, Germany; [12]German Center for Infection Research (DZIF), Partner Site Tübingen, Tübingen, Germany; [13]Institute of Infection, Immunity and Inflammation, University of Glasgow, Glasgow, United Kingdom; [14]Department of Microbiology and Immunology, University of Melbourne, Melbourne, Australia

*For correspondence:
bachmann@bni-hamburg.de

Present address: †GEOMAR Helmholtz Centre for Ocean Research, Kiel, Germany

**Competing interests:** The authors declare that no competing interests exist.

**Abstract** Sequestration of *Plasmodium falciparum*(*P. falciparum*)-infected erythrocytes to host endothelium through the parasite-derived *P. falciparum* erythrocyte membrane protein 1 (*Pf*EMP1) adhesion proteins is central to the development of malaria pathogenesis. *Pf*EMP1 proteins have diversified and expanded to encompass many sequence variants, conferring each parasite a similar array of human endothelial receptor-binding phenotypes. Here, we analyzed RNA-seq profiles of parasites isolated from 32 *P. falciparum*-infected adult travellers returning to Germany. Patients were categorized into either malaria naive (n = 15) or pre-exposed (n = 17), and into severe (n = 8) or non-severe (n = 24) cases. For differential expression analysis, *Pf*EMP1-encoding *var* gene transcripts were de novo assembled from RNA-seq data and, in parallel, *var*-expressed sequence tags were analyzed and used to predict the encoded domain composition of the transcripts. Both approaches showed in concordance that severe malaria was associated with *Pf*EMP1 containing the endothelial protein C receptor (EPCR)-binding CIDRα1 domain, whereas CD36-binding *Pf*EMP1 was linked to non-severe malaria outcomes. First-time infected adults were more likely to develop severe symptoms and tended to be infected for a longer period. Thus, parasites with more pathogenic *Pf*EMP1 variants are more common in patients with a naive immune status, and/or adverse inflammatory host responses to first infections favor the growth of EPCR-binding parasites.

## Introduction

Despite considerable efforts during recent years to combat malaria, the disease remains a major threat to public health in tropical countries. The most severe clinical courses of malaria are due to infections with the protozoan species *Plasmodium falciparum*. In 2019, there were 229 million cases of malaria worldwide, resulting in more than 400,000 deaths (*WHO, 2020*). Currently, about half of the world's population lives in infection-prone areas, and more than 90% of the malaria deaths occur in Africa. In particular, children under five years of age and pregnant women suffer from severe disease, but adults from areas of lower endemicity and non-immune travelers are also vulnerable to severe malaria. Both, children and adults are affected by cerebral malaria, but the prevalence of different features of severe malaria differs with increasing age. Anemia and convulsions are more frequent in children, jaundice indicative of hepatic dysfunction and oliguric renal failure are the dominant manifestations in adults (*Dondorp et al., 2008*; *World Health Organization, 2014*). Moreover, the mortality increases with age (*Dondorp et al., 2008*) and was previously determined as a risk factor for severe malaria and fatal outcome in non-immune patients, but the causing factors are largely unknown (*Schwartz et al., 2001*).

The virulence of *Plasmodium falciparum* (*P. falciparum*) is linked to the infected erythrocytes binding to endothelial cell surface molecules expressed on blood vessel walls. This phenomenon, known as sequestration, prevents the passage of infected erythrocytes through the spleen, which would otherwise remove the infected erythrocytes from the circulation and kill the parasite (*Saul, 1999*). The membrane proteins mediating sequestration are exposed to the host's immune system, and through evolution, *P. falciparum* parasites have acquired several multi-copy gene families coding for variant surface antigens, allowing immune escape through extensive sequence polymorphisms. Endothelial sequestration is mediated by the *P. falciparum* erythrocyte membrane protein 1 (*Pf*EMP1) family, the members of which have different binding capacities for host vascular tissue receptors such as CD36, endothelial protein C receptor (EPCR), ICAM-1, PECAM1, receptor for complement component C1q (gC1qR), and chondroitin sulphate (CSA) (*Magallón-Tejada et al., 2016*; *Turner et al., 2013*; *Rowe et al., 2009*). The long, variable, extracellular *Pf*EMP1 region responsible for receptor binding contains a single N-terminal segment (NTS; main classes A, B, and pam) and a variable number of different Duffy binding-like (DBL; main classes DBLα-ζ and pam) and cysteine-rich inter-domain region domains (CIDR; main classes CIDRα-δ and pam) (*Rask et al., 2010*). These domains were initially allocated to subclasses (e.g., the DBLβ subclasses 1–13); however, due to frequent recombinations between members of the different subclasses, many of these are indistinct and poorly defined (*Otto et al., 2019*).

PfEMP1 molecules have been grouped into four categories (A, B, C, and E) depending on the type of N-terminal domains (the *Pf*EMP1 'head structure') as well as the 5' upstream sequence, the chromosomal localization, and the direction of transcription of their encoding *var* gene (*Rask et al., 2010*; *Kyes et al., 2007*; *Kraemer and Smith, 2003*; *Lavstsen et al., 2003*). Each parasite possesses about 60 *var* genes with approximately the same distribution among the different groups (*Rask et al., 2010*). About 20% of *Pf*EMP1 variants belong to group A and are typically longer proteins with a head structure containing DBLα1 and either an EPCR-binding CIDRα1 domain or a CIDRβ/γ/δ domain of unknown function. Further, the group A includes two conserved, strain-transcendent subfamilies: the *var1* subfamily (previously known as *var1csa*), found in two different variants in the parasite population (3D7- and IT-type) (*Otto et al., 2019*), and the *var3* subfamily, the shortest *var* genes with only two extracellularly exposed domains (DBLα1.3 and DBLε8). Groups B and C include most, about 75%, of *Pf*EMP1 and typically have DBLα0-CIDRα2–6 head structures binding CD36, followed by a DBLδ1-CIDRβ/γ domain combination. A particular subset of B-type proteins, also known as group B/A chimeric genes, possesses a chimeric DBLα0/1 domain (a.k.a. DBLα2) and an EPCR-binding CIDRα1 domain. Thus, the head structure confers mutually exclusive binding properties either to EPCR and CD36, or to an unknown receptor via the CIDRβ/γ/δ domains. C-terminally to the head structure, most group A and some group B and C *Pf*EMP1 have additional DBL domains, of which specific subsets of DBLβ domains of group A and B *Pf*EMP1 bind ICAM-1 (*Lennartz et al., 2017*; *Janes et al., 2011*), and another DBLβ domain, DBLβ12, has been suggested to bind gC1qR (*Magallón-Tejada et al., 2016*). The consistent co-occurrence of specific domain

subsets in the same *Pf*EMP1 gave rise to the definition of domain cassettes (DCs) (*Otto et al., 2019*; *Berger et al., 2013*; *Rask et al., 2010*). The best example of this is the VAR2CSA *Pf*EMP1 (group E, DC2), which binds placental CSA and causes pregnancy-associated malaria (*Salanti et al., 2004*). The VAR2CSA proteins share domain composition, their encoding genes are less diversified than other *var* groups, and all parasites possess one or two *var2csa* copies. Another example is the chimeric group B/A *Pf*EMP1, also known as DC8, which includes DBLα2, specific CIDRα1.1/8 subtypes capable to bind EPCR and typically DBLβ12 domains.

Due to the sequence diversity of *var* genes, studies of *var* expression in patients have relied on the analysis of DBLα-expressed sequence tags (EST) (*Warimwe et al., 2009*; *Warimwe et al., 2012*) informing on relative distribution of different *var* transcripts and qPCR primer sets covering some, but not all, subsets of DBL and CIDR domains (*Lavstsen et al., 2012*; *Mkumbaye et al., 2017*). So far, only very few studies (*Tonkin-Hill et al., 2018*; *Andrade et al., 2020*; *Duffy et al., 2016*; *Kamaliddin et al., 2019*) have used the RNA-seq technology to quantify assembled *var* transcripts in vivo. Moreover, most studies have focused on the role of *Pf*EMP1 in severe pediatric malaria. A consensus from these studies is that severe malaria in children is associated with expression of *Pf*EMP1 with EPCR-binding CIDRα1 domains (*Jespersen et al., 2016*; *Kessler et al., 2017*; *Storm et al., 2019*; *Shabani et al., 2017*; *Mkumbaye et al., 2017*; *Magallón-Tejada et al., 2016*), but elevated expression of dual EPCR and ICAM-1-binding *Pf*EMP1 (*Lennartz et al., 2017*) and the group A-associated DC5 and DC6 have also been associated with severe disease outcomes (*Magallón-Tejada et al., 2016*; *Avril et al., 2013*; *Avril et al., 2012*; *Claessens et al., 2012*; *Lavstsen et al., 2012*; *Duffy et al., 2019*). Less effort has been put into understanding the role of *Pf*EMP1 in relation to severe disease in adults, and its different symptomatology and higher fatality rate. Two gene expression studies from regions of unstable transmission in India showed elevated expression of EPCR-binding variants (DC8, DC13) and DC6 (*Bernabeu et al., 2016*; *Subudhi et al., 2015*), and also of transcripts encoding B- and C-type *Pf*EMP1 in severe cases (*Subudhi et al., 2015*).

In this study, we applied an improved genome-wide expression profiling approach using RNA-seq to study gene expression, in particular *var* gene expression, in *P. falciparum* parasites from hospitalized adult travellers and combined it with a novel prediction analysis of *var* transcripts from DBLα EST. Individuals were clustered into (i) first-time infected (n = 15) and (ii) pre-exposed (n = 17) individuals on the basis of serological data or into (iii) severe (n = 8) and (iv) non-severe (n = 24) cases according to medical reports. Our multi-dimensional analysis revealed a clear association of domain cassettes with EPCR-binding properties with a naive immune status and severe malaria, whereas CD36-binding *Pf*EMP1 proteins and the conserved *var1*-3D7 variant were expressed at higher levels in pre-exposed patients and non-severe cases. Interestingly, severe complications occurred only in the group of first-time infected patients, who also tended to be infected for a longer period, indicating that severity of infection in adults is dependent on duration of infection, host immunity, and parasite virulence gene expression.

## Results

### Cohort characterization

This study is based on a cohort of 32 adult malaria patients hospitalized in Hamburg, Germany. All patients had fever, indicative of symptomatic malaria. MSP1 genotyping estimated a low number of different parasite genotypes present in the patients (*Table 1*). For 10 patients, the present malaria episode was their first recorded *P. falciparum* infection. Nine individuals had previously experienced malaria episodes according to the medical reports, whereas malaria exposure was unknown for 13 patients. In order to determine if patients already had an immune response to *P. falciparum* antigens, indicative of previous exposure to malaria, plasma samples were analyzed by a Luminex-based assay displaying the antigens AMA1, MSP1, and CSP (*Supplementary file 1*). Immune responses to AMA1 and MSP1 are known to be long-lasting, and seroconversion to AMA1 is assumed to occur only after a single or very few infections (*Drakeley et al., 2005*). Principal component analysis (PCA) of the Luminex data resulted in separation of the patients into two discrete groups corresponding to first-time infected adults ('naive cluster') and malaria pre-exposed individuals ('pre-exposed cluster') (*Figure 1A*). The 13 patients with an unknown malaria exposure status could be grouped into either

**Table 1.** Patient data.

| | First-time infected (naive) (n = 15) | Pre-exposed (n = 17) | Severe malaria (n = 8) | Non-severe malaria (n = 24) |
|---|---|---|---|---|
| Female sex [n (%)] | 6 (40%) | 3 (18%) | 3 (38%) | 6 (25%) |
| Patient age in years [median (IQR)] | 34 (26–53) | 38 (31–45) | 47 (27–59) | 35 (31–46) |
| Hb g/dl [median (IQR)]* | 13.1 (12.1–14.6) | 12.2 (11.8–13.1) | 12.1 (11.6–13.0) | 13.2 (12.0–14.3) |
| Parasitaemia % [median (IQR)] | 7.0 (4.0–23.5) | 2.0 (1.0–3.0) | 23.5 (10.0–36.3) | 2.5 (1.0–3.9) |
| Number of MSP1 genotypes [n (%)] | 1: 10 (66%)<br>2: 1 (7%)<br>3: 3 (20%)<br>4: 1 (7%) | 1: 12 (71%)<br>2: 3 (18%)<br>3: 2 (12%)<br>4: 0 (0%) | 1: 5 (63%)<br>2: 1 (13%)<br>3: 1 (13%)<br>4: 1 (13%) | 1: 17 (71%)<br>2: 3 (13%)<br>3: 4 (17%)<br>4: 0 (0%) |
| Total reads [median (IQR)] | 41,341,958 (37,804,417–43,659,324) | 41,259,082 (36,921,362–43,904,892) | 42,458,431 (38,520,154–49,561,881) | 41,050,568 (36,920,201–44,030,863) |
| *P. falciparum* reads [median (IQR)] | 35,940,843 (34,099,395–39,090,313) | 37,065,150 (28,707,096–38,070,441) | 37,980,501 (35,195,959–45,563,701) | 35,559,157 (29,711,534–37,774,576) |
| Number of assembled *var* contigs (>500 bp) [median (IQR)] | 220.5 (169.3–320.8) | 165.5 (121.3–251.5) | 292 (210–404) | 174 (121–259) |
| Parasite age [median (IQR)] | 9.4 (8.0–10.3) | 9.8 (8.0–10.6) | 8.2 (8.0–9.8) | 9.8 (8.2–11.4) |

Geographic origin of the parasite isolates: Ghana (n = 10), Nigeria (n = 6), other Sub-Saharan African countries (n = 15), unknown (n = 1)

*n = 21.

the naive or the pre-exposed group defined by the PCA of antigen reactivity. The only outlier in the clustering was a 19-year-old patient (#21) from Sudan, who reported several malaria episodes during childhood, but clustered with the malaria-naive patients.

Plasma samples were further subjected to (i) a merozoite-directed antibody-dependent respiratory burst (mADRB) assay (*Kapelski et al., 2014*), (ii) a parasitophorous vacuolar membrane-enclosed merozoite structure (PEMS)-specific ELISA, and (iii) a protein microarray with 228 *P. falciparum* antigens (*Borrmann, 2020*). Analysis of these serological assays in relation to the patient clustering confirmed the expected higher and broader antigen recognition by ELISA and protein microarray, and the stronger ability to induce burst of neutrophils by serum from the group of malaria-pre-exposed patients (*Figure 1B–D*, *Supplementary file 1*). Data from all the serological assays were next used for an unsupervised random forest machine learning approach to build models predictive of each individual's protective status. A multidimensional scaling plot was used to visualize cluster allocation, confirming the classification of patient #21 as being non-immune (*Figure 1E*). Patient #26, positioned at the borderline to pre-exposed patients, was grouped into the naive cluster in accordance with the Luminex data and the patient statement that this potentially pre-exposed patient returned from his first trip to Africa. The calculated variable importance highlighted the relevance of mADRB, ELISA, and Luminex assays to allocate patients into clusters (*Figure 1F*).

Using protein microarrays, the antibody response against described antigens was analyzed in detail. As expected, pre-exposed individuals showed significantly elevated IgG antibody responses against a broad range of parasite antigens, especially typical parasite blood stage markers, including MSP1, MSP2, MSP4, MSP10, EBA175, REX1, and AMA1 (*Figure 1D*, upper panel). Markers for a recent infection, MSP1, MSP4, GLURP, and ETRAMP5 (*van den Hoogen et al., 2018*), were significantly elevated in the pre-exposed individuals in comparison to the defined first-time infected group. In addition, further members of the ETRAMP family, including ETRAMP10, ETRAMP14, ETRAMP10.2, and ETRAMP4, and also antibodies against pre-erythrocytic antigens, such as CSP, STARP, and LSA3, were highly elevated. Similar effects were detectable for IgM antibodies; previous exposure to the malaria parasite led to higher antibody levels (*Figure 1D*, lower panel).

Eight patients from the malaria-naive group were considered as having severe malaria based on the predefined criteria. The remaining 24 cases were assigned to the non-severe malaria group (*Figure 1G*, *Supplementary file 2*). Comparing the IgG antibody response of severe and non-severe

A                                                        D

E

                              F                          G

**Figure 1.** Subgrouping of patients into first-time infected and pre-exposed individuals based on antibody levels against *P. falciparum*. In order to further characterize the patient cohort, plasma samples (n = 32) were subjected to Luminex analysis with the *Plasmodium falciparum* (*P. falciparum*) antigens AMA1, MSP1, and CSP, known to induce a strong antibody response in humans. With exception of patient #21, unsupervised clustering of the PCA-reduced data clearly discriminates between first-time infected (naive) and pre-exposed patients with higher antibody levels against tested *P. falciparum* antigens and also assigns plasma samples from patients with an unknown immune status into naive and pre-exposed clusters (**A**). Classification of patient #21 into the naive subgroup was confirmed using different serological assays assessing antibody levels against *P. falciparum* on different levels: a merozoite-directed antibody-dependent respiratory burst (mADRB) assay (*Kapelski et al., 2014*) (**B**), a parasitophorous vacuolar membrane-enclosed merozoite structure (PEMS)-specific ELISA (**C**), and a 262-feature protein microarray covering 228 well-known *P. falciparum* antigens detecting reactivity with individual antigens, and the antibody breadth of IgG (upper panel) and IgM (lower panel) (**D**). The boxes represent medians with IQR; the whiskers depict minimum and maximum values (range), with outliers located outside the whiskers. Serological assays revealed significant differences between patient groups (Mann-Whitney U test). Reactivity of patient plasma IgG and IgM with individual antigens in the protein microarray is presented as the volcano plot, highlighting the significant hits in red. Box plots represent antibody breadths by summarizing the number of recognized antigens out of 262 features tested. Data from all assays were used for an unsupervised random forest approach (**E**). The variable importance plot of the random forest model shows the decrease in prediction accuracy if values of a variable are permuted randomly. The decrease in accuracy was determined for each serological assay, indicating that the mADRB, ELISA, and Luminex assays are most relevant in the prediction of patient clusters (**F**). Venn chart showing the patient subgroups used for differential expression analysis (**G**). Patients with a known immune status based on medical reports were marked in all plots with filled circles in blue (naive) and grey (pre-exposed), and samples from patients with an unknown immune status are shown as open circles. Patient #21 is shown as a filled circle in grey with a cross, and patient #26 is represented by an open circle with a cross. ELISA: enzyme-linked immunosorbent assay; IQR: interquartile range; PCA: principal component analysis.

The online version of this article includes the following figure supplement(s) for figure 1:

**Figure supplement 1.** Early immune response in mild and severe malaria within the naive patient cluster.

cases within the previously malaria-naive group, elevated antibody levels were found in the severe subgroup. The highest fold change was observed for antibodies directed against intracellular proteins, such as DnaJ protein, GTPase-activating protein, or heat shock protein 70 (*Figure 1—figure supplement 1*). Interestingly, IgM antibodies against ETRAMP5 were detectable in the severely infected individuals, suggesting they were infected for a prolonged period of time compared to the mild malaria population (*Helb et al., 2015*; *van den Hoogen et al., 2018*).

## RNA-seq transcriptomics

Parasites were isolated from the venous blood of all patients for subsequent transcriptional profiling (*Figure 2A*). Transcriptome libraries were sequenced for all 32 patient samples (NCBI BioProject ID: PRJNA679547). The number of trimmed reads ranged between 29,142,684 and 82,000,248 (median: 41,383,289) within the individual libraries derived from patients. The proportion of total reads specific for *P. falciparum* was 87.7% (median; IQR 76.7–91.3) for the 30 samples included in the de novo assembly (*Supplementary file 3*). Variation in parasite ages – defined as the progression of the intra-erythrocytic development cycle measured in hours post invasion – in the different patient samples was analyzed with a mixture model in accordance with *Tonkin-Hill et al., 2018*, using the published data from *López-Barragán et al., 2011*. Parasites from first-time infected and pre-exposed patients revealed no obvious difference in the proportion of the different parasite stages or in median age (*Table 1*, *Figure 2—figure supplement 1*). However, a small, statistically not significant bias (p=0.17) towards younger parasites in the severe cases was observed with a median age of 8.2 hours post infection (hpi) (IQR 8.0–9.8) in comparison to 9.8 hpi in the non-severe cases (IQR 8.2–11.4) (*Table 1*, *Figure 2—figure supplement 1D*). None of the samples revealed high proportions of late trophozoites (all <3%), schizonts (0%), or gametocytes (all <6%) (*Figure 2—figure supplement 1A, B*). The estimated proportions were used to control for differences in parasite stages between samples by including them as covariates in the regression analysis of differential core gene expression (*Figure 2A*, *Figure 2—figure supplement 2*).

## Genome-wide analysis of differential gene expression

Global gene expression analysis according to *Tonkin-Hill et al., 2018* identified 420 genes to be higher and 236 to be lower expressed (p≤0.05) in first-time infected patients, together corresponding to 11.3% of all *P. falciparum* genes (*Supplementary file 4*). Similarly, 362 genes were significantly higher and 219 genes lower expressed in severe cases (*Supplementary file 5*). A gene set enrichment analysis (GSEA) of the differentially expressed genes using Gene Ontology (GO) terms and KEGG pathway annotations showed that the KEGG pathway 03410 'base excision repair' facilitating the maintenance of genome integrity by repairing small base lesions in the DNA was expressed at significantly higher levels in first-time infected patient samples (*Figure 2B*, *Figure 2—figure supplement 3*). In total, six out of 15 *P. falciparum* genes included in this KEGG pathway were found to be statistically significantly enriched upon first-time infection, including the putative endonuclease III (PF3D7_0614800) from the short-patch pathway and the putative A-/G-specific adenine glycosylase (PF3D7_1129500), the putative apurinic/apyrimidinic endonuclease Apn1 (PF3D7_1332600), the proliferating cell nuclear antigens 1 (PF3D7_1361900), and the catalytic (PF3D7_1017000) and small (PF3D7_0308000) subunits from the DNA polymerase delta from the long-patch pathway (*Figure 2C*). Additionally, a significantly lower expression level for genes associated with several GO terms involved in antigenic variation and host cell remodeling was found in first-time infected patients (*Figure 2B*, *Supplementary file 4*) and severe cases (*Supplementary file 5*).

As variant surface antigens like *var*, *rif*, and *stevor* are largely clone-specific, analysis of reads from the clinical isolates mapping to homologous regions in 3D7 genes would be distorted and flawed. Therefore, we analyzed *var* gene expression first by de novo assembling *var* genes from the RNA-seq data and subsequently analyzing the expression of specific *var* gene subsets according to the domains encoded. In addition, we manually screened differentially expressed genes known to be involved in *var* gene regulation or the correct display of *Pf*EMP1 at the host cell surface (*Supplementary file 4*, *5*).

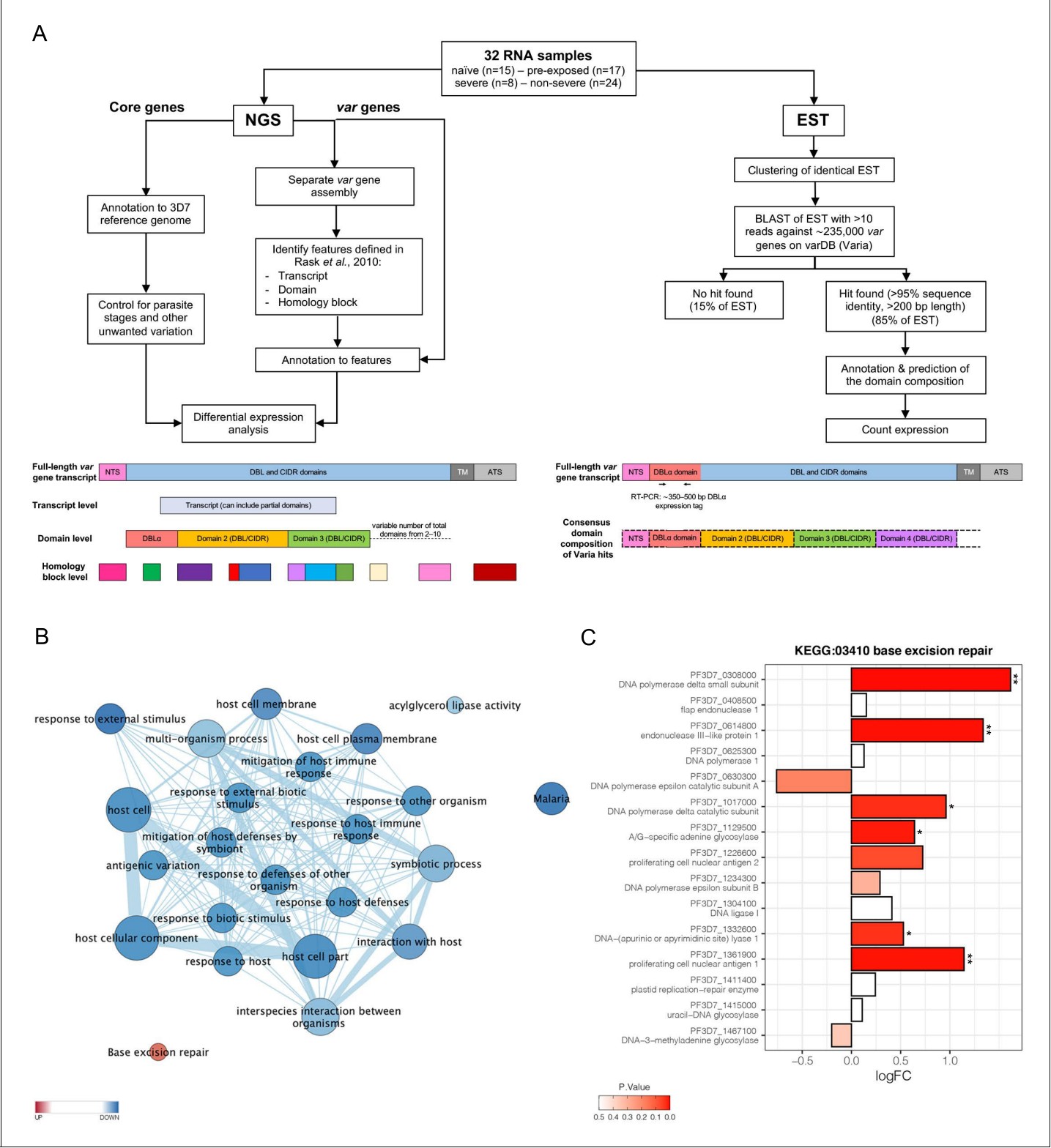

**Figure 2.** Overview of the methodology and differential core gene expression analysis. Summary diagram of the approaches taken to analyze the differential expression of core and *var* genes. In principle, all samples were analyzed by sequencing of the RNA using next generation sequencing (NGS) and by sequencing of expressed sequence tags (EST) from the DBLα domain (**A**). Gene set enrichment analysis (GSEA) of gene ontology (GO) terms and KEGG pathways indicate gene sets deregulated in first-time infected malaria patients. GO terms related to antigenic variation and host cell remodeling are significantly overrepresented in the down-regulated gene set; only the KEGG pathway 03410 'base excision repair' shows a significant

*Figure 2 continued on next page*

*Figure 2 continued*

up-regulation in malaria-naive patients (**B**). Log fold changes (logFC) for the 15 *Plasmodium falciparum* (*P. falciparum*) genes assigned to the KEGG pathway 03410 'base excision repair' are plotted with the six significant hits marked with * p<0.05 and **p<0.01 (**C**).

The online version of this article includes the following figure supplement(s) for figure 2:

**Figure supplement 1.** Estimated stage proportions for each sample.
**Figure supplement 2.** Summary diagram of the approaches taken to analyze the RNA-seq data.
**Figure supplement 3.** The base excision repair (KEGG:03410) in *P. falciparum*.
**Figure supplement 4.** RNA quality.

## Differential *var* gene expression

To correlate individual *var* genes or common *var* gene-encoded traits (*Figure 3*) with a naive immune status or disease severity, differential *var* transcript levels were analyzed as in *Tonkin-Hill et al., 2018*;*Figure 2A*, *Figure 2—figure supplement 2*. *Var* transcripts were assembled from each patient sample separately and annotated. In total, 6441 contigs with over 500 bp lengths were generated with an N50 of 2302 bp and a maximum length of 10,412 bp (*Source data 1.*). A median of 200 contigs (IQR: 137–279) with lengths >500 bp was assembled per sample. One or more DBL or CIDR domains could be annotated to 5488 of the contigs, whereas the remaining contigs could only be annotated by these smaller domain building blocks, the so-called homology blocks as defined by *Rask et al., 2010*; *Supplementary file 6*, *7*.

## Differential *var* transcript levels

We first looked for highly similar transcripts present in multiple samples. The Salmon RNA-seq quantification pipeline (*Patro et al., 2017*), which identifies equivalence classes allowing reads to contribute to the expression estimates of multiple transcripts, was used to estimate expression levels for each transcript. Due to the high diversity in *var* genes, mainly assembled transcripts of the strain-transcendent variants *var1*, *var2csa*, and *var3* were found to be differentially expressed. Notably, the *var1-IT* variant was expressed at higher levels in parasites from first-time infected patients, whereas the *var1-3D7* variant was expressed at higher levels in parasites from pre-exposed and non-severe patients (*Figure 4A,B*, *Figure 5A,B*, *Supplementary file 9*, *10*). This was confirmed by mapping normalized reads from all patients to both *var1* variants as well *as var2csa* (*Figure 3—figure supplement 1*). Beyond the conserved variants, several *var* fragments from B- or C-type *var* genes were associated with a naive immune status, and three transcripts from A-, DC8-, and B-type *var* genes as well as *var2csa* were linked to severe malaria patients (*Figures 4* and *5*, *Supplementary file 9*, *10*).

## Differential *var* domain transcript levels

To assess the differential expression of specific *var* domains, read counts corresponding to domains with the same classification were calculated. This showed that different EPCR-binding CIDRα1 domain variants and other domains found in DC with CIDRα1 domains were expressed at significantly higher levels in first-time infected patients (*Figure 4C–E*, *Supplementary file 9*). Specifically, besides domains from DC8 (DBLα2, CIDRα1.1, DBLβ12) and DC13 (DBLα1.7, CIDRα1.4), the CIDRα1.7 and DBLα1.2 (DC15) domains were increased upon infection of malaria-naive individuals. The DBLα1.2 domain was, in all of the 32 gene assemblies, flanked by an EPCR-binding CIDRα1 domain, and 56% of these were a CIDRα1.5 domain (*Supplementary file 6*, *7*). In addition, parasites from first-time infected patients showed a significantly higher level of transcripts encoding the CIDRδ1 domain of DC16 (DBLα1.5/6-CIDRδ1/2) and the DBLβ6 domain (*Figure 4D–E*). The DBLβ6 domain is associated with A-type *var* genes and can be found adjacent to DC15 and DC16 (*Otto et al., 2019*; *Supplementary file 6*, *7*). In general, domains associated with the same domain cassette showed the same trend even if some domains did not reach statistical significance set at p<0.05 (*Supplementary file 9*).

Domains found expressed at lower levels in malaria-naive individuals included group B and C N-terminal head structure domains NTSB and DBLα0.13/22/23, and CD36-binding CIDR domains (CIDRα2.8/9,6) as well as the C-terminal CIDRγ11 domain and domains of the *var1-3D7* variant (DBLα1.4, DBLγ15, DBLε5) (*Figure 4C–E*).

A

B

**Figure 3.** Summary of *Pf*EMP1 transcripts, domains, and homology blocks that were found more or less frequently in malaria-naive and severely ill patients. A schematic presentation of all *var* gene groups with their associated binding phenotypes and typical *Pf*EMP1 domain compositions. The N-terminal head structure confers mutually exclusive receptor-binding phenotypes: EPCR (beige: CIDRα1.1/4–8), CD36 (turquoise: CIDRα2–6), CSA (yellow: VAR2CSA), and yet unknown phenotypes (brown: CIDRβ/γ/δ; dark red: CIDRα1.2/3 from VAR1, VAR3). Group A includes the conserved subfamilies VAR1 and VAR3, EPCR-binding variants, and those with unknown binding phenotypes conferred by CIDRβ/γ/δ domains. Group B *Pf*EMP1 can have EPCR-binding capacities, but most variants share a four-domain structure, with group C-type variants capable of CD36 binding. Dual binders can be found within groups A and B, with a DBLβ domain after the first CIDR domain responsible for ICAM-1 (DBLβ1/3/5) or gC1qr binding (DBLβ12) (**A**). Transcripts, domains, and homology blocks according to *Rask et al., 2010* as well as domain predictions from the DBLα-tag approach were found to be significantly differently expressed (p-value<0.05) between patient groups of both comparisons: first-time infected (blue) versus pre-exposed (black) cases and severe (red) versus non-severe (black) cases (**B**). ATS: acidic terminal sequence; CIDR: cysteine-rich interdomain region; CSA: chondroitin sulphate A; DBL: Duffy binding-like; DC: domain cassette; EPCR: endothelial protein C receptor; gC1qr: receptor for complement

*Figure 3 continued on next page*

*Figure 3 continued*

component C1q; ICAM-1: intercellular adhesion molecule 1; NTS: N-terminal segment; PAM: pregnancy-associated malaria; TM: transmembrane domain.

The online version of this article includes the following figure supplement(s) for figure 3:

**Figure supplement 1.** Differential expression of the *var1* variants 3D7 and IT and *var2csa* between patient groups.

When comparing severe to the non-severe cases, domains of DC8 (DBLα2, CIDRα1.1, DBLβ12), DC15 (DBLα1.2), and DC16 (DBLα1.6) and the A-type-linked DBLβ6 were found associated with severe disease. Domain types expressed at significantly higher levels in non-severe cases included N-terminal head structure domains, DBLα0.23, CIDRα2.4/9 from group B and C *Pf*EMP1, DC16 (DBLα1.5), and the CIDRα1.3 domain from the *var1-3D7* allele (*Figure 5C–E*, *Supplementary file 10*).

As DBL and CIDR subclasses are poorly defined (*Otto et al., 2019*) and different domain subclasses confer the same binding phenotype (*Higgins and Carrington, 2014*), the domains of the N-terminal head structure were grouped according to their binding phenotype and the normalized read counts (transcripts per million [TPM]) were summarized for each patient (*Figures 4F* and *5F*). This showed significant differences for domains associated with EPCR- or CD36-binding *Pf*EMP1. As expected, domains associated with EPCR-binding as well as the CIDRγ3 domain were expressed at higher levels in naive and more severe cases, whereas domains associated with the CD36-binding phenotype were higher expressed in pre-exposed and non-severe patients.

## Differential *var* homology block transcript levels

*Pf*EMP1 domains have been described as composed of 628 homology blocks (*Rask et al., 2010*). Homology block expression levels were obtained by aggregating read counts for each block after first identifying all occurrences of the block within the assembled contigs. Transcripts encoding block numbers 255, 584, and 614, all typically located within DBLβ domains of DC8- and CIDRα1-containing type A *Pf*EMP1 (*Figure 4G,H*, *Supplementary file 9*), number 557, located in the interdomain region between DBLβ and DBLγ domains (no *Pf*EMP1 type association), and block number 155, found in NTSA, were found associated with a naive immune status. Conversely, transcripts encoding block number 88 from DBLα0 domains and 269 from ATSB were found at lower levels in malaria-naive patients, indicating that B- and C-type genes are more frequently expressed in pre-exposed individuals (*Figure 4G,H*, *Supplementary file 9*). No homology blocks were associated with severe cases, but two blocks, 591 and 559, found in group B *Pf*EMP1, were found to be lower expressed in severe malaria cases (*Figure 5G,H*, *Supplementary file 10*).

## *Var* expression profiling by DBLα-tag sequencing

To supplement the RNA-seq analysis with an orthogonal analysis, we conducted deep sequencing of RT-PCR-amplified DBLα ESTs from 30 of the patient samples (*Lavstsen et al., 2012*; *Figure 2A*). Between 851 and 3368 reads with a median of 1666 over all samples were analyzed. Identical DBLα-tag sequences were clustered to generate relative expression levels of each unique *var* gene tag. Overall, the relative expression levels were similar for sequences found in both the RNA-seq and the DBLα-tag approach with a mean log2(DBLα-PCR/RNA-seq) ratio of 0.4 (CI of 95%: −2.5–3.3) determined by Bland-Altman plotting (*Figure 6—figure supplement 1*). Around 82.6% (median) of all unique DBLα-tag sequences detected with >10 reads (92.9% of all DBLα-tag sequences) were found in the RNA-seq approach, and 81.8% of the upper 75th percentile of RNA-seq contigs (spanning across the DBLα-tag region) were found by the DBLα-tag approach.

Using the Varia tool (*Mackenzie et al., 2020*), the domain composition of the *var* genes from which the DBLα-tag sequences originated was predicted. The tool searches an extended varDB containing >200K annotated *var* genes for genes with near identical DBLα-tag sequences and returns the consensus domain annotation among the hit sequences. A partial domain annotation was made for ~85% of the DBLα-tag sequences (*Figure 2A*, *Supplementary file 11*). In line with the RNA-seq data, this analysis showed that DBLα1 and DBLα2 sequences were enriched in first-time infected and severe malaria patients. Conversely, a significant higher proportion of DBLα0 sequences was found in pre-exposed individuals and mild cases (*Figure 6A,B*). No difference was observed

E

H

**Figure 4.** Expression differences between parasites from first-time infected and pre-exposed patients at the level of *var* gene transcripts, domains, and homology blocks determined by NGS. RNA-seq reads of each patient sample were matched to de novo assembled *var* contigs with varying lengths, domains, and homology block compositions. Shown are significantly differently expressed *var* gene contigs (**A, B**) as well as *Pf*EMP1 domain subfamilies (**C–F**) and homology blocks (**G, H**) from *Rask et al., 2010*, with an adjusted p-value of <0.05. Data are displayed as heat maps showing expression

*Figure 4 continued on next page*

*Figure 4 continued*

levels either in log-transformed normalized Salmon read counts (A) or in log transcripts per million (TPM) (C, G) for each individual sample. Box plots show median log-transformed normalized Salmon read counts (B) or TPM (D, F, H) and interquartile range (IQR) for each group of samples. Individual domains from inter-strain conserved tandem arrangements of domains, so-called domain cassettes (DCs), found significantly higher expressed in samples from first-time infected (blue arrow) and pre-exposed patients (grey arrow), are indicated in bold (E). The N-terminal head structure (NTS-DBLα-CIDRα/β/γ/δ) confers a mutually exclusive binding phenotype either to EPCR-, CD36-, CSA-, or an unknown receptor. Expression values of the N-terminal domains were summarized for each patient, and differences in the distribution among patient groups were tested using the Mann-Whitney U test (F). Normalized Salmon read counts for all assembled transcripts and TPM for *Pf*EMP1 domains and homology blocks are available in *Supplementary file 8*.

in the number of reads or unique DBLα-tags detected between patient groups, although a trend towards more DBLα-tag clusters was observed in first-time infected patients and severe cases (*Figure 6A,B*). Prediction of the NTS and CIDR domains flanking the DBLα domain showed a significantly higher proportion of NTSA in severe cases as well as EPCR-binding CIDRα1 domains in first-time infected and severe cases. Expression of *var* genes encoding NTSB and CIDRα2–6 domains was significantly associated within pre-exposed and non-severe cases (*Figure 6A,B*). Subsequent analysis of *var* expression in relation to other domains showed that *var* transcripts with DBLβ, γ, and ζ or CIDRγ domains were more frequently expressed in first-time infected and severe malaria patients whereas those encoding DBLδ and CIDRβ were less frequent (*Figure 6—figure supplement 2*). Assessing expression in relation to the domain subtype, CIDRα1.1/5, DBLβ12, DBLγ2/12, DBLα2, DBLα1.2/2, and DBLδ5 (together with DBLγ12 components of the DC5) were found associated with severe malaria, while CIDRα3.1/4, DBLα0.12/16, and DBLδ1 were associated with non-severe cases (*Figure 6—figure supplement 2*).

Overall, these data corroborate the main observations from the RNA-seq analysis, confirming the association of EPCR-binding *Pf*EMP1 variants with development of severe malaria symptoms and CD36-binding *Pf*EMP1 variants with establishment of less severe infections in semi-immune individuals.

## Correlation of *var* gene expression with antibody levels against head structure CIDR domains

A detailed analysis of the antibody repertoire of the patients against head structure CIDR domains of *Pf*EMP1 was carried out using a panel of 19 different EPCR-binding CIDRα1 domains, 12 CD36-binding CIDRα2–6 domains, three CIDRδ1 domains as well as a single CIDRγ3 domain (*Obeng-Adjei et al., 2020*; *Bachmann et al., 2019*). Additionally, the minimal binding region of VAR2CSA was included. In general, plasma samples from malaria-naive as well as severe cases showed lower mean fluorescence intensity (MFI) values for all antigens tested in comparison to samples from pre-exposed or non-severe cases with significant differences for CIDRα2–6, CIDRδ1, and CIDRγ3, but not for EPCR-binding CIDRα1 domains (*Figure 7A,B*).

Data were also analyzed using the average MFI reactivity (plus two standard deviations) of a Danish control cohort as a cut off for seropositivity to calculate the coverage of antigen recognition (*Supplementary file 1*; *Cham et al., 2010*). In this analysis, almost half of the tested antigens were recognized by pre-exposed (median: 47.2%) and non-severe patients (median: 44.4%), but only 1/4 of the antigens were recognized by first-time infected patients (median: 25.0%) and 1/20 by severely ill patients (median: 5.6%). *Pf*EMP1 antigens recognized by over 60% of the pre-exposed and/or non-severe patient sera were (i) four CIDRα1 domains capable of EPCR-binding (CIDRα1.5, CIDRα1.6, CIDRα1.7, and the DC8 domain CIDRα1.8), (ii) two CD36-binding CIDRα domains (CIDRα2.10, CIDRα3.1), and (iii) two CIDR domains with unknown binding phenotypes (CIDRδ1 and CIDRγ3) (*Figure 7C*, *Supplementary file 1*).

Taken together, this analysis indicates that higher levels of antibodies against severe malaria-associated EPCR-binding variants are present in pre-exposed and non-severe cases, which might have selected against parasites expressing CIDRα1 domains during the current infection.

C

E

D

**Figure 5.** Expression differences between parasites from severe and non-severe cases at the level of *var* gene transcripts, domains, and homology blocks determined by NGS. RNA-seq reads of each patient sample were matched to de novo assembled *var* contigs with varying lengths, domains, and homology block compositions. Shown are significantly differently expressed *var* gene contigs (**A, B**) as well as *Pf*EMP1 domain subfamilies (**C–F**) and homology blocks from *Rask et al., 2010*, with an adjusted p-value of <0.05 in severe (red) and non-severe patient samples (grey) (**A, B**). Data are
*Figure 5 continued on next page*

*Figure 5 continued*

displayed as heat maps showing expression levels either in log-transformed normalized Salmon read counts (**A**) or in log transcripts per million (TPM) (**C, G**) for each individual sample. Box plots show median log-transformed normalized Salmon read counts (**B**) or TPM (**D, F, H**) and interquartile range (IQR) for each group of samples. Individual domains from inter-strain conserved tandem arrangements of domains, so-called domain cassettes (DCs), found significantly higher expressed in severe (red arrow) and non-severe cases (grey arrow), are indicated in bold (**E**). The N-terminal head structure (NTS-DBLα-CIDRα/β/γ/δ) confers a mutually exclusive binding phenotype either to EPCR-, CD36-, CSA-, or an unknown receptor. Expression values of the N-terminal domains were summarized for each patient, and differences in the distribution among patient groups were tested using the Mann-Whitney U test (**F**). Normalized Salmon read counts for all assembled transcripts and TPM for *Pf*EMP1 domains and homology blocks are available in *Supplementary file 8*.

## Discussion

Non-immune travellers and adults from areas of unstable malaria transmission are prone to severe malaria. Currently, only scarce information on the *Pf*EMP1-mediated pathogenicity responsible for the different symptomatology in comparison to paediatric severe malaria and the higher fatality rate in adults is available. Here, we present the first in-depth gene expression analysis of 32 ex vivo blood samples from adult travellers using RNA-seq and expressed sequence tag analyses. Despite the relatively low number of patient samples recruited in 5 years, our data confirmed previously reported associations between transcripts encoding type A and B EPCR-binding *Pf*EMP1 and infections in naive hosts and disease severity (*Duffy et al., 2019*; *Tonkin-Hill et al., 2018*; *Kessler et al., 2017*; *Bernabeu et al., 2016*; *Jespersen et al., 2016*; *Lavstsen et al., 2012*). Our results further suggest that parasite interaction with EPCR is linked to severe disease in children as well as in adults. However, since CIDRα1-containing *Pf*EMP1 possess multiple binding traits (*Lennartz et al., 2017*; *Magallón-Tejada et al., 2016*), co-interaction with other receptors may further increase the risk for severe malaria.

Overall, there was a high degree of consensus between observations made on *var* expression analyzed at different levels of *Pf*EMP1 domain annotation. Stratifying *var* gene expression according to the different main and subtypes of DBL and CIDR domains showed only A- and DC8-type *Pf*EMP1 domains, and predominantly those linked to EPCR-binding *Pf*EMP1, to be associated with first-time infections. Conversely, domains typical for CD36-binding *Pf*EMP1 proteins were found at higher levels in malaria-experienced adults. Specifically, expression of *Pf*EMP1 domains included in DC8, DC13, and DC15 as well as all EPCR-binding CIDRα1 domains was associated with first-time adult infections, whereas DBLα0- and CD36-binding CIDRα2–6 domains were linked to pre-exposed individuals. These differences were largely due to the differential expression between the first-time infected patients with more severe symptoms and patients with non-severe malaria. Here, domains of DC8 and DC15 as well as all DBLα1/2 and CIDRα1were associated with severe symptoms, while NTSB, DBLα0, and CIDRα2–6 domains, including specific subsets of CIDRα2, were linked to non-severe symptoms. These conclusions were closely mirrored in the DBLα-tag analysis and were further corroborated by the differential RNA-seq expression stratified according to the smaller homology blocks, which identified mainly homology blocks of DBLβ1, -3, -5, and -12 to be associated with first-time infected patients. These DBLβ domains are parts of DCs associated with EPCR binding; so it is hard to distinguish between co-occurring domains and clear associations.

In addition, three other group A *Pf*EMP1-associated domains, CIDRγ3, CIDRδ from DC16, DBLβ9 from DC5, and DBLβ6 were found to be associated with first-time infected patients. DBLβ9 and DBLβ6 could have been detected due to their presence C-terminally to some EPCR-binding or A-type *Pf*EMP1. However, the CIDRδ domain of DC16 (DBLα1.5/6-CIDRδ1/2) constitutes a different subset of A-type *Pf*EMP1, which, together with CIDRβ2- and CIDRγ3-containing group A *Pf*EMP1 (found in DC11), may be associated with rosetting (*Carlson et al., 1990*; *Ghumra et al., 2012*). Direct evidence that any of these CIDR domains have intrinsic rosetting properties is lacking (*Rowe et al., 2002*). Rather, their association with rosetting may be related to their tandem expression with DBLα1 at the N-terminal head (*Ghumra et al., 2012*). The DC16 group A signature was not associated with severe disease outcomes in previous DBLα-tag studies or qPCR studies by *Lavstsen et al., 2012* and *Bernabeu et al., 2016*, but DBLα1.5/6 and CIDRδ of DC16 were enriched in cerebral malaria cases with retinopathy in the study of *Shabani et al., 2017* and *Kessler et al., 2017* using the same qPCR primer set. Also, the association of DC11 with severe malaria in Indonesia was found using the same RNA-seq approach as used here (*Tonkin-Hill et al., 2018*). Rosetting is

A

**Figure 6.** Verification of RNA-seq results using DBLα-tag sequencing. Amplified DBLα-tag sequences were blasted against the ~2400 genomes on varDB (*Otto, 2019*) to obtain subclassification into DBLα0/1/2 and prediction of adjacent head structure N-terminal segment (NTS) and cysteine-rich interdomain region (CIDR) domains and their related binding phenotype. Proportion of each NTS and DBLα subclass as well as CIDR domains grouped according to the binding phenotype (CIDRα1.1/4–8: EPCR-binding, CIDRα2–6: CD36-binding, CIDRβ/γ/δ: unknown binding phenotype/rosetting) was calculated and shown separately on the left, and the number of total reads and individual sequence cluster with n $\geq$ 10 sequences are shown on the right. Differences in the distribution among first-time infected (blue) and pre-exposed individuals (grey) (**A**) as well as severe (red) and non-severe cases (grey) (**B**) were tested using the Mann-Whitney U test. The boxes represent medians with IQR; the whiskers depict minimum and maximum values (range) with outliers located outside the whiskers.

The online version of this article includes the following figure supplement(s) for figure 6:

**Figure supplement 1.** Comparison of DBLα-tag sequencing with RNA-seq analysis.

**Figure supplement 2.** Quantile regression analysis of Varia outputs.

C

**Figure 7.** Correlation of *var* gene expression with antibody levels against head structure CIDR domains. Patient plasma samples (n = 32) were subjected to Luminex analysis with 35 *Pf*EMP1 head structure cysteine-rich interdomain region (CIDR) domains. The panel includes endothelial protein C receptor (EPCR)-binding CIDRα1 domains (n = 19), CD36-binding CIDRα2–6 domains (n = 12), and CIDR domains with unknown binding phenotypes (CIDRγ3: n = 1, CIDRδ1: n = 3) as well as the minimal binding region of VAR2CSA (VAR2). Box plots showing mean fluorescence intensities (MFI) extending from the 25th to the 75th percentile with a line at the median indicate the higher reactivity of the pre-exposed (**A**) and non-severe cases (**B**) with all *Pf*EMP1 domains tested. Significant differences were observed for recognition of CIDRα2–6, CIDRδ1, and CIDRγ3; VAR2CSA recognition differed only between severe and non-severe cases (Mann-Whitney U test). Furthermore, the breadth of IgG recognition (%) of CIDR domains for the different patient groups was calculated and shown as a heat map (**C**).

infection in children with severe malaria, severe anemia, and cerebral malaria, and transcripts with CIDRα2–6 domains are most abundantly expressed during uncomplicated malaria (*Jespersen et al., 2016*; *Duffy et al., 2019*; *Warimwe et al., 2009*; *Warimwe et al., 2012*). Although the median expression of CIDRα2–6 is lower in first-time infected and severe cases compared to pre-exposed and non-severe cases, in most of our adult patients, CD36-binding *var* transcripts appear to dominate the expression pattern. This is in concordance with all three other adult studies also indicating a substantial expression of B- and C-type variants associated with the binding of CD36 (*Argy et al., 2017*; *Bernabeu et al., 2016*; *Subudhi et al., 2015*), and (*Subudhi et al., 2015*) even showed an association with complicated adult malaria. Maybe parasite binding to CD36 is specifically enhanced in adult severe malaria cases compared to children, which is interesting due to their different disease symptomatology (*Dondorp et al., 2008*; *Schwartz et al., 2001*). Alternatively, our adult cohort differs not only in age but also in terms of disease severity from paediatric cohorts, and less sick patients may simply have a less dominant expression of EPCR-binding variants. However, for the parasite's survival and transmission, it may be highly beneficial to express more of the less virulent *Pf*EMP1 variants that are able to bind CD36. This interaction may not, or is less likely to, result in obstruction of blood flow, inflammation, and organ failure at least of the brain, where CD36 is nearly absent (*Turner et al., 1994*).

To the best of our knowledge, this study is the first description of expression differences between the two *var1* variants, 3D7 and IT. The *var1-IT* variant was found enriched in parasites from first-time infected patients, whereas several transcripts of the *var1*-3D7 variant were increased in pre-exposed and non-severely ill patients. Expression of *var1* gene was previously observed to be elevated in malaria cases imported to France with an uncomplicated disease phenotype (*Argy et al., 2017*). In general, the *var1* subfamily is ubiquitously transcribed (*Winter et al., 2003*; *Duffy et al., 2006*), atypically late in the cell cycle after transcription of *var* genes encoding the adhesion phenotype (*Kyes et al., 2003*; *Duffy et al., 2002*; *Dahlbäck et al., 2007*) and is annotated as a pseudogene in 3D7 due to a premature stop codon. Similarly, numerous isolates display frame-shift mutations often in exon two in the full gene sequences (*Rask et al., 2010*). However, none of these studies addressed the differences in the two *var1* variants that were recently identified by comparing *var* gene sequences from 714 *P. falciparum* genomes (*Otto et al., 2019*), and to date, it is still unclear if both variants fulfill the same function or have the same characteristics as previously described. Overall, the *var1* gene – the first 3.2 kb of the 3D7 variant in particular – seems to be under high evolutionary pressure (*Otto et al., 2019*), and both variants can be traced back before the split of *P. reichenowi* from *P. praefalciparum* and *P. falciparum* (*Otto et al., 2018b*). Our data indicate that the two variants, VAR1-3D7 and VAR1-IT, may have different roles during disease; however, this remains to be determined in future studies.

In summary, our data show a significant increase in transcripts encoding EPCR-binding and other A-type variants in parasites from severe and first-time infected patients. Conversely, transcripts of CD36-binding variants are found more frequently in parasites from non-severe and pre-exposed patients. Since CD36-binding variants are still overrepresented in all groups of adult malaria patients, we postulate that the parasite population in first-time infected individuals may have a broad binding potential after liver release as there is no pre-existing immunity to clear previously experienced *Pf*EMP1 variants. During the blood-stage infection, selection towards EPCR binding and other A-type variants, which may confer a parasite growth advantage and also increase the risk for severe malaria, may already have occurred in our adult severe malaria patients as indicated by the longer period of infection.

## Materials and methods

### Blood sampling and processing

EDTA blood samples (1–30 mL) were obtained from the adult patients. The plasma was separated by centrifugation and immediately stored at −20℃. Erythrocytes were isolated by Ficoll gradient centrifugation followed by filtration through Plasmodipur filters (EuroProxima) to clear the remaining granulocytes. An aliquot of erythrocytes (about 50–100 µl) was separated and further processed for gDNA purification. At least 400 µl of purified erythrocytes were rapidly lysed in five volumes of pre-warmed TRIzol (ThermoFisher Scientific) and stored at −80℃ until further processing.

## Serological assays

### Luminex assay

The Luminex assay was conducted as previously described using the same plex of antigens tested (*Bachmann et al., 2019*). In brief, plasma samples from patients were screened for the individual recognition of 19 different CIDRα1, 12 CIDRα2–6, three CIDRδ1 domains, and a single CIDRγ3 domain as well as of the controls AMA1, MSP1, CSP, VAR2CSA (VAR2), tetanus toxin (TetTox), and bovine serum albumin (BSA). The data are shown as MFI, allowing comparison between different plasma samples, but not between different antigens. Alternatively, the breadth of antibody recognition (%) was calculated using MFI values from Danish controls plus two standard deviations (SD) as cut off.

### Merozoite-triggered antibody-dependent respiratory burst

The assay to determine the mADRB activity of the patients was set up as described before (*Kapelski et al., 2014*). Polymorphonuclear neutrophil granulocytes (PMNs) from one healthy volunteer were isolated by a combination of dextran-sedimentation and Ficoll-gradient centrifugation. Meanwhile, $1.25 \times 10^6$ merozoites were incubated with 50 µl of 1:5 diluted plasma (decomplemented) from adult patients as well as from established negative and positive control donors for 2 hr. The opsonized merozoites were pelleted (20 min, 1500 g), re-suspended in 25 µl Hanks' Balanced Salt Solution (HBSS), and then transferred to a previously blocked well of an opaque 96-well high-binding plate (Greiner Bio-One). Chemiluminescence was detected in HBSS using 83.3 µM luminol and $1.5 \times 10^5$ PMNs at 37°C for 1 hr to characterize the PMN response, with readings taken at 2-min intervals using a multiplate reader (CLARIOstar, BMG Labtech). PMNs were added in the dark, immediately before readings were initiated.

### ELISA

Antibody reactivity against PEMS was estimated by ELISA. PEMS were isolated as described before (*Llewellyn et al., 2015*). For ELISA, $0.625 \times 10^5$ PEMS were coated on the ELISA plates in PBS. Plates were blocked using 1% Casein (Thermo Scientific #37528) and incubated for 2 hr at 37°C. After washing using phosphate-buffered saline (PBS)/0.1% Tween, plasma samples from patients and control donors were added at two-fold dilutions of 1:200 to 1:12,800 in PBS/0.1% casein. The samples were incubated for 2 hr at room temperature (RT). IgG was quantified using HRP-conjugated goat anti-human IgG at a dilution of 1:20,000 and incubated for 1 hr. For the colour reaction, 50 µl of TMB substrate was used and stopped by adding 1 M HCl after 20 min. Absorbance was quantified at 450 nm using a multiplate reader (CLARIOstar, BMG Labtech).

### Protein microarray

Microarrays were produced at the University of California Irvine, Irvine, California, USA (*Doolan et al., 2008*). In total, 262 *P. falciparum* proteins representing 228 unique antigens were expressed using an *Escherichia coli* (*E. coli*) lysate in vitro expression system and spotted on a 16-pad ONCYTE AVID slide. The selected *P. falciparum* antigens are known to frequently provide a positive signal when tested with sera from individuals with sterile and naturally acquired immunity against the parasite (*Obiero et al., 2019*; *Dent et al., 2016*; *Doolan et al., 2008*; *Felgner et al., 2013*). For the detection of binding antibodies, secondary IgG antibody (goat anti-human IgG QDot800; Grace Bio-Labs #110635), secondary IgM antibody (biotin-SP-conjugated goat anti-human IgM; Jackson ImmunoResearch #109-065-043), and Qdot585 Streptavidin Conjugate (Invitrogen #Q10111MP) were used (*Taghavian et al., 2018*).

Study serum samples as well as the positive and European control sera were diluted 1:50 in 0.05X Super G Blocking Buffer (Grace Bio-Labs, Inc) containing 10% *E. coli* lysate (GenScript, Piscataway, NJ) and incubated for 30 min on a shaker at RT. Meanwhile, microarray slides were rehydrated using 0.05X Super G Blocking buffer at RT. Rehydration buffer was subsequently removed and samples added onto the slides. Arrays were incubated overnight at 4°C on a shaker (180 rpm). Serum samples were removed the following day and microarrays were washed using 1X TBST buffer (Grace Bio-Labs, Inc). Secondary antibodies were then applied at a dilution of 1:200 and incubated for 2 hr at RT on the shaker, followed by another washing step and a 1 hr incubation in a 1:250 dilution of Qdot585 Streptavidin Conjugate. After a final washing step, slides were dried by centrifugation at

500 g for 10 min. Slide images were taken using the ArrayCAM Imaging System (Grace Bio-Labs) and the ArrayCAM 400 s Microarray Imager Software.

Microarray data were analyzed in R statistical software package version 3.6.2. All images were manually checked for any noise signal. Each antigen spot signal was corrected for local background reactivity by applying a normal-exponential convolution model (*McGee and Chen, 2006*) using the RMA-75 algorithm for parameter estimation (available in the LIMMA package v3.28.14) (*Silver et al., 2009*). Data was log2-transformed and further normalized by subtraction of the median signal intensity of mock expression spots on the particular array to correct for background activity of antibodies binding to *E. coli* lysate. After log2 transformation, data approached normal distribution. Differential antibody levels (protein array signal) in the different patient groups were determined by Welch-corrected Student's t-test. Antigens with p<0.05 and a fold change >2 of mean signal intensities were defined as differentially recognized between the tested sample groups. Volcano plots were generated using the PAA package (*Turewicz et al., 2016*) and GraphPad Prism 8. Individual antibody breadths were defined as the number of seropositive features with signal intensities exceeding an antigen-specific threshold set at six standard deviations above the mean intensity in negative control samples.

### Unsupervised random forest model

An unsupervised random forest (RF) model, a machine learning method based on multiple classification and regression trees, was calculated to estimate proximity between patients. Variable importance was calculated, which shows the decrease in prediction accuracy if values of a variable are permuted randomly. The k-medoids clustering method was applied on the proximity matrix to group patients according to their serological profile. Input data for random forest were Luminex measurements for MSP1, AMA1, and CSP reduced by principal component analysis (PCA; first principal component selected), mADRB, and ELISA, and antibody breadths of IgG and IgM determined by protein microarray were used to fit the RF model. Multidimensional scaling was used to display patient cluster. All analyses were done with R (4.02) using the packages randomForest (4.6–14) to run RF models and cluster (2.1.0) for k-medoids clustering.

## Patient classification according to severity

Severity was defined in line with the WHO criteria for severe malaria in adults (World Health Organization (WHO), 2014). Patients were considered as having severe malaria if they showed signs of impaired organ function (e.g., jaundice, renal failure, and cerebral manifestations) or had extremely high parasitaemia (>10%). In addition, patients #1 and #26 were included into the severe group due to circulating schizonts indicative of a very high sequestering parasite biomass associated with severity (*Bernabeu et al., 2016*; *Supplementary file 2*).

## DNA purification and MSP1 genotyping

Genomic DNA was isolated using the QIAamp DNA Mini Kit (Qiagen) according to the manufacturer's protocol. To assess the number of *P. falciparum* genotypes present in the patient isolates, MSP1 genotyping was carried out as described elsewhere (*Robert et al., 1996*).

## RNA extraction, RNA-seq library preparation, and sequencing

TRIzol samples were thawed, mixed rigorously with 0.2 volumes of cold chloroform,and incubated for 3 min at RT. After centrifugation for 30 min at 4°C and the maximum speed, the supernatant was carefully transferred to a new tube and mixed with an equal volume of 70% ethanol. Subsequently, the manufacturer's instructions from the RNeasy MinElute Kit (Qiagen) were followed with DNase digestion (DNase I, Qiagen) for 30 min on column. Elution of the RNA was carried out in 14 µl. Human globin mRNA was depleted from all samples except from samples #1 and #2 using the GLOBINclear kit (ThermoFisher Scientific). The quality of the RNA was assessed using the Agilent 6000 Pico kit with Bioanalyzer 2100 (Agilent), and the RNA quantity was assessed using the Qubit RNA HA assay kit and a Qubit 3.0 fluorometer (ThermoFisher Scientific). Upon arrival at BGI Genomics Co (Hong Kong), the RNA quality of each sample was double-checked before sequencing. The median RIN value over all ex vivo samples was 6.75 (IQR: 5.93–7.40) (*Figure 2—figure supplement 4*), although this measurement has only limited significance for samples containing RNA of two species.

Customized library construction in accordance with *Tonkin-Hill et al., 2018*, including amplification with KAPA polymerase and HiSeq 2500 100 bp paired-end sequencing, was also performed by BGI Genomics Co (Hong Kong).

## RNA-seq read mapping and data analysis

### Differential expression of core genes

Differential gene expression analysis of *P. falciparum* core genes was done in accordance with *Tonkin-Hill et al., 2018*), using the scripts available in the GitHub repository (https://github.com/gtonkinhill/falciparum_transcriptome_manuscript/tree/master/all_gene_analysis). In brief, subread-align v1.4.6 (*Liao et al., 2013*) was used to align the reads to the *Homo sapiens* (*H. sapiens*) and *P. falciparum* reference genomes. Read counts for each gene were obtained with FeatureCounts v1.20.2 (*Liao et al., 2014*). To account for parasite life cycle, each sample was considered as a composition of six parasite life cycle stages, excluding the ookinete stage (*López-Barragán et al., 2011*). Unwanted variations were determined with the 'RUV' (Remove Unwanted Variation) algorithm implemented in the R package ruv v0.9.6 (*Gagnon-Bartsch and Speed, 2012*) adjusting for systematic errors of unknown origin by using the genes with the 1009 lowest p-values as controls as described in *Vignali et al., 2011*. The gene counts and estimated ring-stage factor, and factors of unwanted variation were then used as inputs for the Limma/Voom (*Law et al., 2014*; *Smyth, 2005*) differential analysis pipeline.

### Functional enrichment analysis of differentially expressed core genes

Genes that were identified as significantly differentially expressed (defined as $-1 < \text{logFC} > 1$, $p<0.05$) during prior differential gene expression analysis were used for functional enrichment analysis using the R package gprofiler2 (*Kolberg et al., 2020*). Enrichment analysis was performed on multiple input lists containing genes expressed significantly higher (logFC >1, $p<0.05$) and lower (logFC <-1, $p<0.05$) between different patient cohorts. All *var* genes were excluded from the enrichment analysis. For custom visualization of results, gene set data sources available for *P. falciparum* were downloaded from gprofiler (*Raudvere et al., 2019*). Pathway data available in the KEGG database (https://www.kegg.jp/kegg/) was accessed via the KEGG API using KEGGREST (*Tenenbaum, 2020*) to supplement gprofiler data sources and build a custom data source in Gene Matrix Transposed file format (*.gmt) for subsequent visualization. Functional enrichment results were then output to a Generic Enrichment Map (GEM) for visualization using the Cytoscape EnrichmentMap app (*Merico et al., 2010*) and RCy3 (*Gustavsen et al., 2019*). Bar plots of differential gene expression values for genes of selected KEGG pathways were generated using ggplot2 (*Wickham, 2016*) and enriched KEGG pathways were visualized using KEGGprofile (*Zhao et al., 2020*).

## *Var* gene assembly

Samples from patient #1 and #2 not subjected to globin-mRNA depletion due to their low RNA content after multiple rounds of DNase treatment showed low percentages of *P. falciparum*-specific reads (12.4% and 15.68%) (*Supplementary file 2*). Consequently, less than one million *P. falciparum* reads were obtained for each of these samples, and they were omitted from assembly due to low coverage, but included in the differential gene expression analysis.

*Var* genes were assembled using the pipeline described in *Tonkin-Hill et al., 2018*. The separate assembly approach was chosen since it reduces the risk for generating false chimeric genes and results in longer contigs compared to the combined all sample assembly approach. Briefly, non-*var* reads were first filtered out by removing reads that aligned to *H. sapiens*, *P. vivax,* or non-*var P. falciparum*. Assembly of the remaining reads was then performed using a pipeline combining SOAPdenovo-Trans and Cap3 (*Xie et al., 2014*; *Huang and Madan, 1999*; *Liao et al., 2013*). Finally, contaminants were removed from the resulting contigs and they were then translated into the correct reading frame. Reads were mapped to the contigs using BWA-MEM (*Li, 2013*) and RPKM values were calculated for each *var* transcript to compare individual transcript levels in each patient. Although transcripts might be differentially covered by RNA-seq due to their variable GC content, this seems not to be an issue between *var* genes (*Tonkin-Hill et al., 2018*).

## *Var* transcript differential expression

Expression of the assembled *var* genes was quantified using Salmon v0.14.1 (*Patro et al., 2017*) for 531 transcripts with five read counts in at least three patient isolates. Both the naive and pre-exposed groups as well as the severe and non-severe groups were compared. The combined set of all de novo assembled transcripts was used as a reference. As the RNA-seq reads from each sample were assembled independently it is possible for a highly similar transcript to be present multiple times in the combined set of transcripts from all samples. The Salmon algorithm identifies equivalence sets between transcripts, allowing a single read to support the expression of multiple transcripts. As a result, Salmon accounts for the redundancy present in our whole set of *var* gene contigs from all separate sample-specific assemblies. To confirm the suitability of this approach, we also ran the Corset algorithm as used in *Tonkin-Hill et al., 2018* and *Davidson and Oshlack, 2014*. Unlike Salmon, which attempts to quantify the expression of transcripts themselves, Corset copes with the redundancy present in de novo transcriptome assemblies by clustering similar transcripts together using both the sequence identity of the transcripts as well as multi-mapping read alignments. Of the transcripts identified using the Salmon analysis, 5/15 in the naive versus pre-exposed and 4/13 in the severe versus non-severe were identified in the significant clusters produced using Corset. As the two algorithms take very different approaches and as Salmon is quantifying transcripts rather than the 'gene' like clusters of Corset, this represents a fairly reasonable level of concordance between the two methods. However, due to the high diversity in *var* genes, both of these approaches are only able to identify significant associations between transcripts and phenotypes when there is sufficient similarity within the associated sequences. In both the Salmon and Corset pipelines, differential expression analysis of the resulting *var* expression values was performed using DESeq2 v1.26 (*Love et al., 2014*). The Benjamini-Hochberg method was used to control for multiple testing (*Benjamini and Hochberg, 1995*).

To check the differential expression of the conserved *var* gene variants *var1*-3D7, *var1*-IT, and *var2csa*, raw reads were mapped with BWA-MEM (AS score >110) to the reference genes from the 3D7 and the IT strains. The mapped raw read counts (bam files) were normalized with the number of 3D7 mappable reads in each isolate using bamCoverage by introducing a scaling factor to generate bigwig files displayed in Artemis (*Carver et al., 2012*).

## *Var* domain and homology block differential expression

Differential expression analysis at the domain and homology level was performed using a similar approach to that described previously (*Tonkin-Hill et al., 2018*). Initially, the domain families and homology blocks defined in Rask et al. were annotated to the assembled transcripts using HMMER v3.1b2 (*Rask et al., 2010*; *Eddy, 2011*). Domains and homology blocks previously identified to be significantly associated with severe disease in *Tonkin-Hill et al., 2018*) were also annotated by single pairwise comparison in the assembled transcripts using USEARCH v11.0.667 (*Tonkin-Hill et al., 2018*; *Edgar, 2010*). Overall, 336 contigs (5.22% of all contigs > 500 bp) possess partial domains in an unusual order, for example, an NTS in an internal region or a tandem arrangement of two DBLα or CIDRα domains. This might be caused by de novo assembly errors, which is challenging from transcriptome data. Therefore, in both cases, the domain or homology block with the most significant alignment was taken as the best annotation for each region of the assembled *var* transcripts (E-value cut off of $1e^-8$). The expression at each of these annotations was then quantified using featureCounts v1.6.4 before the counts were aggregated to give a total for each domain and homology block family in each sample. Finally, similar to the transcript level analysis, DESeq2 was used to test for differences in expression levels of both domain and homology block families in the naive versus pre-exposed groups as well as the severe versus non-severe groups. Again, more than five read counts in at least three patient isolates were required for inclusion into differential expression analysis.

## DBLα-tag sequencing

For DBLα-tag PCR, the forward primer varF_dg2 (5'-tcgtcggcagcgtcagatgtgtataagagacagGCAMG MAGTTTYGCNGATATWGG-3') and the reverse primer brlong2 (5'-gtctcgtgggctcggagatgtgtataaga-gacagTCTTCDSYCCATTCVTCRAACCA-3') were used resulting in an amplicon size of 350–500 bp (median 422 bp) plus the 67 bp overhang (small type). Template cDNA (1 µl) was mixed with 5x KAPA HiFi buffer, 0.3 µM of each dNTP, 2 µM of each primer and 0.5 U KAPA HiFi Hotstart

Polymerase in a final reaction volume of 25 µl. Reaction mixtures were incubated at 95℃ for 2 min and then subjected to 35 cycles of 98℃ for 20 s, 54℃ for 30 s, and 68℃ for 75 s with a final elonga-tion step at 72℃ for 2 min. For the first five cycles, cooling from denaturation temperature was per-formed to 65℃ at a maximal ramp of 3℃ per second, then cooled to 54℃ with a 0.5℃ per second ramp. Heating from annealing temperature to elongation temperature was performed with 1℃ per second, all other steps with a ramp of 3℃ per second. Agarose gel images taken afterwards showed clean amplicons. The DBLα-tag primers contain an overhang, which was used to conduct a second indexing PCR reaction using sample-specific indexing primers as described in *Nag et al., 2017*. The overhang sequence also serves as annealing site for Illumina sequencing primers, and indexing pri-mers include individual 8-base combinations and adapter sequences that will allow the final PCR product to bind in MiSeq Illumina sequencing flow cells. Indexing PCR reactions were performed with a final primer concentration of 0.065 µM and 1 µl of first PCR amplicon in a final volume of 20 µl using the following steps: heat activation at 95℃, 15 min, 20 cycles of 95℃ for 20 s, 60℃ for 1 min and 72℃ for 1 min, and one final elongation step at 72℃ for 10 min. Indexing PCR amplicons were pooled (4 µl of each) and purified using AMPure XP beads (Beckman Coulter, California, United States) according to the manufacturer's protocol, using 200 µl pooled PCR product and 0.6 x PCR-pool volume of beads, to eliminate primer dimers. The purified PCR pools were analyzed on agarose gels and Agilent 2100 Bioanalyser to verify the elimination of primer dimers and correct amplicon sizes. Concentration of purified PCR pools was measured by Nanodrop2000 (Thermo Fisher Scien-tific, Waltham, MA, USA), and an aliquot adjusted to 4 nM concentration was pooled with another unrelated DNA material and added to an Illumina MiSeq instrument for paired-end 300 bp reads using a MiSeq v3 flow cell.

## DBLα-tag sequence analysis

The paired-end DBLα-tag sequences were identified and partitioned into the correct sample origin based on unique index sequences. Each indexed raw sequence-pair was then processed through the *Galaxy* webtool (usegalaxy.eu). Read quality checks were first performed with *FastQC* to ensure a good NGS run (sufficient base quality, read length, duplication etc.). Next, the sequences were trimmed by the Trimmomatic application, with a four-base sliding window approach and a *Phred* quality score above 20 to ensure high sequence quality output. The trimmed sequences were then paired and converted, following analysis using the Varia tool for quantification and prediction of the domain composition of the full-length *var* sequences from which the DBLα-tag originated (*Mackenzie et al., 2020*). In brief, Varia clusters DBLα-tags with 99% sequence identity using *Vsearch* program (v2.14.2), and each unique tag is used to search a database consisting of roughly 235,000 annotated *var* genes for near identical *var* sequences (95% identity over 200 nucleotides). The domain composition of all 'hit' sequences is checked for conflicting annotations, and the most likely domain composition is retuned. The tool validation indicated prediction of correct domain compositions for around 85% of randomly selected *var* tags, with a higher hit rate and accuracy of the N-terminal domains. An average of 2,223.70 reads per patient sample was obtained, and clus-ters consisting of less than 10 reads were excluded from the analysis. The raw Varia output file is given in *Supplementary file 10*. The proportion of transcripts encoding a given *Pf*EMP1 domain type or subtype was calculated for each patient. These expression levels were used to first test the hypothesis that N-terminal domain types associated with EPCR are found more frequently in first-time infections or upon severity of disease, while those associated with non-EPCR binding were asso-ciated with pre-exposed or mild cases. Secondly, quantile regression was used to calculate median differences (with 95% confidence intervals) in expression levels for all main domain classes and sub-types between severity and exposure groups. All analyses were done with R (4.02) using the package quantreg (5.73) for quantile regression.

For the comparison of both approaches, DBLα-tag sequencing and RNA-seq, only RNA-seq con-tigs spanning the whole DBLα-tag region were considered. All conserved variants, the subfamilies *var1*, *var2csa,* and *var3*, detected by RNA-seq, were omitted from analysis since they were not prop-erly amplified by the DBLα-tag primers. To scan for the occurrence of DBLα-tag sequences within the contigs assembled from the RNA-seq data, we applied BLAST (basic local alignment search tool) v2.9.0 software (*Altschul et al., 1990*). Therefore, we created a BLAST database from the RNA-seq assemblies and screened for the occurrence of those DBLα-tag sequence with more than 97% per-cent sequence identity using the 'megablast option'.

Calculation of the proportion of RNA-seq data covered by DBLα-tag was done with the upper 75th percentile based on total RPKM values determined for each patient. Vice versa, only DBLα-tag clusters with more than 10 reads were considered and percent coverage of reads and clusters calculated for each individual patient.

For all samples, the agreement between the two molecular methods DBLα-tag sequencing and RNA-seq was analyzed with a Bland-Altman plot, each individually, and summarized. The ratios between %-transformed measurements are plotted on the y-axis and the mean of the respective DBLα-tag and RNA-seq results is plotted on the x-axis. The bias and the 95% limits of agreement were calculated using GraphPad Prism 8.4.2.

## Acknowledgements

We thank all the patients who provided an extra blood sample for our research purposes. We would also like to thank the staff of the I Medical Department at the UKE and of the Bundeswehrkrankenhaus for identifying patients for the study and, specifically, Maria Mackroth, Julian Schulze zur Wiesch, Johannes Jochum, and Thierry Rolling for assisting in the recruitment of patients. Furthermore, we thank Jürgen May for critical reading of the manuscript and Tobias Spielmann for helpful discussions. We thank Marlene Danner Dalgaard, Kathrine Hald Langhoff, and Sif Ravn Søeborg for technical assistance with DBLα-tag sequencing.

## Additional information

### Funding

| Funder | Grant reference number | Author |
| --- | --- | --- |
| Deutsche Forschungsgemeinschaft | BA 5213/3-1 | Jan Stephan Wichers<br>Anna Bachmann |
| Danish Council for Independent Research | 9039-00285B | Rasmus Weisel Jensen<br>Louise Turner<br>Thomas Lavstsen |
| Deutsches Zentrum für Infektionsforschung | TTU Malaria | Ralf Krumkamp<br>Egbert Tannich<br>Rolf Fendel<br>Anna Bachmann |
| DESY | PIF-2018-87 | Jan Strauss<br>Tim Wolf Gilberger |
| Universität Hamburg | | Judith Anna Marie Scholz |
| National Health and Medical Research Council | | Michael Duffy |
| Kirsten og Freddy Johansens Fond | | Rasmus Weisel Jensen<br>Louise Turner<br>Thomas Lavstsen |
| Lundbeckfonden | R344-2020-934 | Rasmus Weisel Jensen<br>Louise Turner<br>Thomas Lavstsen |
| Wellcome Trust | 104111/Z/14/ZR | Thomas D Otto |

The funders had no role in study design, data collection and interpretation, or the decision to submit the work for publication.

### Author contributions

J Stephan Wichers, Formal analysis, Investigation, Writing - original draft; Gerry Tonkin-Hill, Formal analysis, Investigation, Visualization, Methodology, Writing - review and editing; Thorsten Thye, Formal analysis, Methodology, Writing - review and editing; Ralf Krumkamp, Formal analysis, Investigation, Visualization, Methodology; Benno Kreuels, Writing - review and editing, Patient recruitment; Jan Strauss, Formal analysis, Visualization, Writing - review and editing; Heidrun von Thien, Judith

AM Scholz, Investigation; Helle Smedegaard Hansson, Rasmus Weisel Jensen, Louise Turner, Freia-Raphaella Lorenz, Anna Schöllhorn, Investigation, Methodology; Iris Bruchhaus, Data curation; Egbert Tannich, Conceptualization, Patient recruitment; Rolf Fendel, Investigation, Visualization, Methodology, Writing - review and editing; Thomas D Otto, Conceptualization, Data curation, Formal analysis, Investigation, Methodology, Writing - review and editing; Thomas Lavstsen, Conceptualization, Investigation, Visualization, Methodology, Writing - original draft; Tim W Gilberger, Supervision, Writing - review and editing; Michael F Duffy, Conceptualization, Methodology, Writing - review and editing; Anna Bachmann, Conceptualization, Formal analysis, Supervision, Funding acquisition, Investigation, Visualization, Methodology, Writing - original draft, Project administration

## Author ORCIDs

J Stephan Wichers https://orcid.org/0000-0002-0599-1742
Gerry Tonkin-Hill http://orcid.org/0000-0003-4397-2224
Thorsten Thye https://orcid.org/0000-0002-8720-7113
Ralf Krumkamp https://orcid.org/0000-0003-3053-476X
Benno Kreuels https://orcid.org/0000-0003-2315-8954
Jan Strauss http://orcid.org/0000-0002-6208-791X
Helle Smedegaard Hansson https://orcid.org/0000-0001-6484-1165
Freia-Raphaella Lorenz http://orcid.org/0000-0002-7401-2720
Anna Schöllhorn http://orcid.org/0000-0001-7928-7312
Iris Bruchhaus http://orcid.org/0000-0002-3363-7409
Egbert Tannich http://orcid.org/0000-0002-4714-7275
Rolf Fendel http://orcid.org/0000-0003-4716-0311
Thomas D Otto https://orcid.org/0000-0002-1246-7404
Thomas Lavstsen http://orcid.org/0000-0002-3044-4249
Tim W Gilberger https://orcid.org/0000-0002-7965-8272
Anna Bachmann https://orcid.org/0000-0001-8397-7308

## Ethics

Human subjects: The study was conducted according to the principles of the Declaration of Helsinki in its 6th revision as well as International Conference on Harmonization-Good Clinical Practice (ICH-GCP) guidelines. Blood samples for this analysis were collected after patients were informed about the aims and risks of the study and signed an informed consent form for voluntary blood draw (n=21). In the remaining cases, no designated blood samples were drawn, instead remains from diag-nostic blood samples were used (n=11). The study was approved by the relevant ethics committee (Ethical Review Board of the Medical Association of Hamburg, reference numbers PV3828 and PV4539).

## Decision letter and Author response

Decision letter https://doi.org/10.7554/eLife.69040.sa1
Author response https://doi.org/10.7554/eLife.69040.sa2

---

# Additional files

## Supplementary files

• Source data 1. Sequences from all assembled *var* contigs with a length >500 bp.

• Supplementary file 1. Data from Luminex, mADRB, ELISA, and protein microarray. Seroprevalence of head structure CIDR domains determined by applying a cut off from Danish controls (mean +2 STD) to the Luminex data.

• Supplementary file 2. Characteristics and classification of adult malaria patients. Parasitaemia, signs of organ failure, and subgrouping of each individual patient.

• Supplementary file 3. Raw read counts by sample for *H. sapiens*, *P. falciparum*, *var* exon one and percentage of reads that mapped either to *P. falciparum* or *var* exon one as well as the number of assembled *var* contigs > 500 bp in length.

• Supplementary file 4. Differentially expressed genes excluding *var* genes (all gene analyses) between first-time infected and pre-exposed patient samples.

• Supplementary file 5. Differentially expressed genes excluding *var* genes (all gene analyses) between severe and non-severe patient samples.

• Supplementary file 6. Features of the assembled *var* fragments annotated in accordance with *Rask et al., 2010* and *Tonkin-Hill et al., 2018*. The reading frame used for translation is given after the contig ID, and the position of each annotation is provided by starting and ending amino acid followed by the p-value from the blast search against the respective database. For annotations in accordance with *Tonkin-Hill et al., 2018*, either the short ID or 'NA' (not applicable) is listed at the end. Short IDs are only available for significant differently expressed domains and blocks between severe and non-severe cases (*Tonkin-Hill et al., 2018*).

• Supplementary file 7. Summary of *var* gene fragments assembled for each patient isolate showing length, raw read counts, RPKM, blast hits, domains, and block annotations in accordance with *Rask et al., 2010*. The RPKM for the contigs was calculated as the number of mapped reads and normalized by the number of mapped reads against all transcripts in each isolate, respectively. Therefore, RPKM expression values are only valid for comparison within a single sample since RNA-seq reads were mapped only to the contigs of the respective patient isolate using BWA-MEM (*Li, 2013*). Further, the amount of blast hits with 500 bp or 80% of overlap against the ~2400 samples from varDB (*Otto, 2019*) with an identity cut off of 98%. Further hits of 1 kb (>98% identity) against the *var* genes from the 15 reference genomes (*Otto et al., 2018a*) are listed. The last two columns show the annotations from *Rask et al., 2010* associated to each contig.

• Supplementary file 8. Log-transformed normalized Salmon read counts for assembled *var* transcripts, TPM for collapsed domains, and homology blocks from each patient isolate. Normalized counts and transcripts per million (TPM) values calculated for transcripts, domains, and blocks with expression in at least three patient isolates with more than five read counts.

• Supplementary file 9. Differently expressed *var* transcripts, domains, and homology blocks between first-time infected and pre-exposed patient samples.

• Supplementary file 10. Differently expressed *var* transcripts, domains, and homology blocks between severe and non-severe patient samples.

• Supplementary file 11. Data from DBLα-tag sequencing.

• Transparent reporting form

## Data availability

Sequencing data have been deposited at NCBI under the BioProject accession number PRJNA679547.

The following dataset was generated:

| Author(s) | Year | Dataset title | Dataset URL | Database and Identifier |
|---|---|---|---|---|
| Wichers JS, Tonkin-Hill G, Thye T, Krumkamp R, Kreuels B, Strauss J, Thien H, Scholz JAM, Hansson SH, Jensen WR, Turner L, Lorenz F-R, Schöllhorn A, Bruchhaus I, Tannich E, Fendel R, Otto TD, Lavstsen T, Gilberger T-W, | 2020 | Data from: Common virulence gene expression in adult first-time infected patients and severe cases | https://www.ncbi.nlm. nih.gov/bioproject/ PRJNA679547 | NCBI BioProject, PRJNA679547 |

Duffy MF,
Bachmann A

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
