## [Decision Letter]

[Editors' note: this paper was reviewed by Review Commons.]

**Acceptance summary:**

This study describes a detailed analysis of gene expression in *Plasmodium falciparum*, malaria causing parasite, isolated from a group of adult travelers returning to Germany. The authors developed new approaches that allowed them to confirm the presence of particular parasite variants, whereby one is associated with severe malaria and the other with non-severe malaria infection. This is a comprehensive study with an interesting dataset and the findings highlight the key knowledge gap about variant antigenic expression in adult severe malaria.

---

## [Author Response]

Reviewer #1Summary: The submission from Wichers and colleagues describes a detailed analysis of gene expression in *P. falciparum* malaria parasites isolated from a cohort of patients in Germany who recently traveled in regions of the world with significant malaria transmission. The authors analyzed gene expression in parasites isolated from 32 patients and associated specific expression profiles with disease severity and pre-exposure of the patient to malaria. In particular, the authors focused on var/PfEMP1 expression since this gene family has long been associated with disease severity. The overall findings of the study are largely supportive of previous conclusions from other researchers and reinforce the hypothesis that PfEMP1s that bind to EPCR are associated with severe disease while those binding to CD36 are typically expressed in non-severe disease. The researchers also report a shift in the "age" of circulating parasites between severe and non-severe disease as well as a curious association with expression of specific var1 alleles.Major comments:1. The authors detected a shift in "age" of parasites obtained from non-severe infections compared to severe malaria patients, which they thought was important enough to include the in the abstract of the paper. However, in the Results section, the authors refer to this as a "small bias" and it seems that the difference (as displayed in Figure 2) does not reach statistical significance. The shift in "age" might indeed be important and therefore is worth pointing out to the readers, however the authors should explicitly state that the difference does not reach statistical significance. Given that it is mentioned prominently in the abstract, it is important not to mislead readers.

We decided to omit the statement about parasite age in the abstract and moved figure 2 into the supplements (Figure 2 —figure supplement 1). Additionally, we included a statement in the manuscript that the observed difference in age did not reach statistical significance (Line 215ff): “However, a small, statistically not significant bias (p=0.17) towards younger parasites in the severe cases was observed with a median age of 8.2 hpi (IQR: 8.0–9.8) in comparison to 9.8 hpi in the non-severe cases (IQR: 8.2–11.4) (Table 1, Figure 2 —figure supplement 1D).”

2. The authors performed differential expression analysis using their RNA-seq datasets and identified genes that change in expression according to disease severity and the patient's exposure status. However, given that the authors also detected a shift in "age" (or more specifically stage of the asexual cycle) between the different sample subsets, these changes in gene expression could simply reflect the shifted cycle. A reanalysis of the gene sets that shift up or down in the different samples should enable the authors to determine if these shifts can be explained simply by a shift in timing. If the change in timing explains many or most of the changes in gene expression, this provides additional support for the authors' conclusion that parasite "age" correlates with disease severity. In contrast, genes in which the change in expression does not correlate with the shift in "age" might be directly involved in disease severity.

We currently do control for the parasite age in the differential gene expression analysis by including the inferred parasite stage proportions as covariates in differential expression regression analysis. However, we realized this could be made clearer in the text. We have now added the following sentence to clarify that the differential expression analysis accounts for the lifecycle stage of the parasite within each sample (Line 220ff): “The estimated proportions were used to control for differences in parasite stage between samples by including them as covariates in the regression analysis of differential gene expression (Figure 2A, Figure 2 —figure supplement 2).” Moreover, we have inserted an overview of the methods used in Figure 2A and Figure 2 —figure supplement 2 showing the control for parasite stages and other unwanted factors of variation in the analysis of core gene expression.

Minor comments:3. The authors refer to the "age" of the parasites in their samples, by which they mean how far into the 48 hour asexual cycle the parasites have progressed. This is fine, however readers not intimately familiar with this concept might find this initially confusing. It would be good to define what the authors mean by age when they first mention this term.

Thanks for pointing this out. We included a definition of ‘parasite age’ in Line 209ff: “Variation in parasite ages – defined as the progression of the intraerythrocytic development cycle measured in hours post invasion – in the different patient samples was analyzed with a mixture model in accordance to Tonkin-Hill et al. (Tonkin-Hill et al., 2018) using published data from López-Barragán et al. as a reference (López-Barragán et al., 2011).”

4. The authors focus somewhat on patient #21 in analysis of the data shown in Figure 1. It might be helpful to mark on the panels in the figure which of the circles represent this specific patient. If I looked close, I could figure it out, but it would be more intuitive if it were marked, or example with an arrow or asterisk.

We have marked patient #21 and #26 in all panels of Figure 1 as indicated in the figure legend (Line 1171f): “Patient #21 is shown as filled circle in grey with a cross, patient #26 is represented by an open circle with cross.”

5. Also in Figure 1, the authors assay for previous exposure to *P. falciparum* by measuring several immune responses. I was a bit surprised that they didn't include a few samples from uninfected individuals who have never traveled to regions of malaria transmission to serve as negative controls.

We actually included negative controls in all serological assays and positive controls in mADRB, ELISA and protein microarray. This data is provided in the Supplementary table S1. Furthermore, we now explicitly state that also in the Material and Method section:

– For the mADRB assay (Line 546ff): “*Meanwhile, 1.25 x 10^6^ merozoites were incubated with 50 µl of 1:5 diluted plasma (decomplemented) from adult patients as well as from established negative and positive control donors for 2 h.*”

– For ELISA (Line 559f): “*After washing using PBS/0.1% Tween, plasma samples from patients and control donors were added at two-fold dilutions of 1:200 to 1:12800 in PBS/0.1% Casein*.”

– For protein microarray (Line 578ff): “*Study serum samples as well as the positive and European control sera were diluted 1:50 in 0.05X Super G Blocking Buffer (Grace Bio-Labs, Inc.) containing 10% E. coli lysate (GenScript, Piscataway, NJ) and incubated for 30 minutes on a shaker at RT.*”

We also would like to point out that a Danish cohort was also used to determine the background MFI in the Luminex assay and served as a cut-off (plus 2 STD) for seropositivity in Figure 7C. The data set for the whole danish cohort as well as the calculation of the threshold for seropositivity can be found in Supplementary Table S1. This is mentioned in the text line 374ff: “*The data was also analyzed using the average MFI reactivity (plus two standard deviations) of an Danish control cohort as a cut off for seropositivity to calculate the coverage of antigen recognition (Table S1) (Cham et al., 2010).*” And in Line 538ff: “*Alternatively, the breadth of antibody recognition (%) was calculated using MFI values from Danish controls plus two standard deviations*

*(SD) as cut off.*”

We feel that addition of more sample groups in the plots of Figure 1 would make this already data-dense figure even more complex. Additionally, since the individuals used for control samples differed between the serological assays (in dependence of the location the assays were performed), the controls cannot be integrated into the random forest approach.

6. In the analysis shown in Figure 4, the authors note that var1, var2csa and var3 are expressed at higher frequencies in first-time infected patients. It would be good to mark the contigs representing these genes on the figure.

We have grouped and marked contigs of the conserved *var* gene variants *var1*, *var2csa* and *var3* in Figure 4 and 5 to make the figures more readable. See also major comment 7 and minor comments of Reviewer 2.

7. The authors refer to var1 when describing the conserved gene found within one of the subtelomeric regions of chromosome 5. Since this gene was originally named "var1csa", it would be good for the authors to mention the original name when first describing the gene to avoid confusion. I support the idea of renaming this gene to simply var1 since the "csa" part of the name is misleading, however explicitly stating that they are referring to this previously named gene will be helpful.

We referred to the original name of the *var1* variant by placing “*previously known as var1csa*” into brackets in Line 94ff when we first mention the gene.

8. The tables listed at the end of the manuscript (Tables 1-8) were not available. I suspect this was an error in how the manuscript files were uploaded.

Sorry for that. Yes, that was indeed an error during the upload process, which has already been corrected as recognized. Since we removed Table 1 and 3-8 and included the data into the main figures, the updated version contains only a single table (previous Table 2).

Significance: Overall this is a comprehensive and detailed study that researchers studying malaria pathogenesis will find valuable. Some of the specifics, for example the nuances of the different combinations of PfEMP1 domains, might appeal to a somewhat narrow readership, although the methods and approaches that the authors employed are broadly valuable.My expertise: Gene expression analysis of malaria parasites.Referees cross-commenting: Similar to reviewer #3, I was also happy to see that the three reviews are in agreement that no additional experiments are requested. The authors should be able to modify the manuscript with the reviewers’ comments in mind and make the paper suitable for publication.Reviewer #2Summary: The var gene family is a major virulence determinant for *P. falciparum*, but it is extremely challenging to study because of the large size of genes and the diversity of var genes between parasite genotypes. This study developed new approaches to analyze and classify var genes expressed by patient isolates. The technical approaches are state of the art and provide the most complete picture to date of the types of parasite variants that are circulating in patients with different malaria symptoms. The authors provide strong evidence that the ratio of different subsets of parasite variants (EPCR and CD36 binding) are skewed in patient subsets with elevated EPCR-binding var transcripts in malaria naïve and severe malaria patients and elevated CD36-binding variants in non-severe infections. The new approaches for var gene profiling and classification of predicted binding phenotypes are a major technical advance. Conversely, there is no direct evidence that the EPCR-binding phenotype confers more efficient sequestration and alternative hypotheses have not been considered. Overall, the new methodology is a tour de force, but the manuscript has not been written in a way that will be easily understood by a broad readership. I have some suggestions to improve its clarity.Major comments:

*1. A strength of the manuscript is the detailed characterization of parasite var transcripts by both NGS analysis and deep sequencing of DBLa amplicons. These two approaches were leveraged by the powerful bioinformatic classification into different functional groupings. The conclusion about skewed expression of EPCR-binding var transcripts in severe malaria and CD36-binding var transcripts in non-severe patients is strongly supported by the combined var profiling approaches*.

We are grateful for this comment.

2. At the same time, it is worthwhile to point out that all patients are infected by a mixture of parasite variants and that CD36-binding var transcripts appear to be the dominant var transcripts in most patients, median ~60% in firsttime infected and ~80% in pre-exposed (Figure 7). I think that the literature has been more focused on the association of EPCR with severity, but less attention has been placed on the proportion of different var subsets in all patients.

Indeed, most studies – including this one – are looking at differentially expressed genes since measurements of total abundances of the different *var* groups are even more challenging. Previous studies tried to estimate transcript abundance encoding CD36-binding variants were hampered by a lower coverage or poor sensitivity of qPCR primer pairs targeting these N-terminal head structure domains (DBLa0, CIDRa2-6). Vice versa, DBLatag approaches are slightly biased towards CD36-binding variants (recognize more DBLa0 than DBLa1/2) and fail to detect most of the conserved *var* variants. Therefore, the abundance of certain transcripts may have been underestimated in the different approaches. However, there is cumulative evidence also from other studies, that CIDRa1 containing transcripts are dominating the infection in children with severe malaria, severe anemia and cerebral malaria. In contrast, transcripts with CIDRa2-6 domains are most abundantly expressed during uncomplicated malaria (Jespersen *et al.*, 2016; Duffy *et al.*, 2019; Warimwe *et al.*, 2009 & 2012). Our adult patient cohort differs not only in age of the patients, but also in terms of severity, so maybe our firsttime infected and less sick adult patients have less dominant expression of EPCR-binding variants. This is also in concordance with the studies by Bernabeu *et al.* (2016) and Subudhi *et.al.* (2015) who analyzed *var* expression in adult Indian cohorts. Using either qPCR primers targeting mainly domain cassettes or a custom cross-strain microarray they also found an elevated expression of domains with EPCR, rosetting and CD36 binding PfEMP1 in severe adult cases. This may indicate that parasite binding to CD36 is enhanced in adult severe malaria cases compared to children, which is interesting due to their different disease symptomatology.

To address this in the manuscript the following paragraph was inserted into the Discussion section (Line 457ff): “Measurements of the total abundances of the different var groups are challenging (e.g., due differences in the coverage or sensitivity of qPCR or DBLα-tag primer pairs targeting the different var groups), but there is cumulative evidence that CIDRα1 containing transcripts are dominating the infection in children with severe malaria, severe anemia and cerebral malaria and transcripts with CIDRα2-6 domains are most abundantly expressed during uncomplicated malaria (Jespersen et al., 2016; Duffy et al., 2019; Warimwe et al., 2009, 2012). Although the median expression of CIDRα2-6 is lower in first-time infected and severe cases compared to preexposed and non-severe cases, in most of our adult patients CD36-binding var transcripts appear to dominate the expression pattern. This is in concordance with all three other adult studies also indicating a substantial expression of B- and C-type variants associated with binding of CD36 (Argy et al., 2017; Bernabeu et al., 2016; Subudhi et al., 2015) and Subudhi et al. even show an association with complicated adult malaria (Subudhi et al., 2015). Maybe parasite binding to CD36 is specifically enhanced in adult severe malaria cases compared to children, which is interesting due to their different disease symptomatology (Dondorp et al., 2008; Schwartz et al., 2001). Alternatively, our adult cohort differs not only in age but also in terms of disease severity from pedatric cohorts, and less sick patients may simply have a less dominant expression of EPCR-binding variants. However, for the parasite’s survival and transmission, it may be highly beneficial to express more of the less virulent PfEMP1 variants able to bind CD36. This interaction may not, or is less likely to, result in obstruction of blood flow, inflammation and organ failure at least of the brain, where CD36 is nearly absent (Turner et al., 1994).”

3. There is no direct evidence that patients with first-time infection were selected "for parasites with more efficient sequestration" (Line 44) or "that the EPCR-binding phenotype confers more efficient sequestration of infected erythrocytes" (Line 127). Indeed, predicted CD36 var transcripts are highly expressed in all patients, so differences in circulating parasite ages are difficult to attribute to any given cytoadhesion trait. Moreover, alternative hypothesis for earlier sequestration have not been considered, such as febrile temperature increases cytoadherence of ring-infected P. falciparum-infected erythrocytes (Udomsanpetch et al. PNAS 2002, PMID 12177447) or that parasites may respond to host factors, like lactate levels, by increasing other transcripts that may modify the IE rigidity or adhesion strength (e.g. Lee et al. Nat. Micro. 2019, PMID 29950443 or their own previous study Tonkin-Hill Plos Biology 2018). Although the age of circulating parasites was earlier in severe cases, it did not reach significance (Figure 2D). Unless there is more direct evidence for this conclusion, it should be removed from the abstract and qualified, as well as consider other hypotheses that could contribute to the strength of ringIE cytoadherence.

We agree, that no direct evidence is given and hence we modified/deleted the statements mentioned by the reviewer in the abstract (Line 45ff) and in the last paragraph of the introduction (Line 141ff) accordingly. The paragraph concerning the earlier sequestration upon severity has been removed from the discussion and we modified the last paragraph (Line 496ff). Figure 2 showing the stage distribution has been moved to the Supplement (Figure 2 —figure supplement 1).

– Abstract (Line 45ff): “First-time infected adults were more likely to develop severe symptoms and tended to be infected for a longer period. Thus, parasites with more pathogenic PfEMP1 variants are more common in patients with a naïve immune status and/or adverse inflammatory host responses to first infections favors growth of EPCR-binding parasites.”

– Introduction (Line 141ff): “Interestingly, severe complications occurred only in the group of first-time infected patients who tended to be infected for a longer period, indicating that severity of infection in adults is dependent on duration of infection, host immunity and parasite virulence gene expression.”

– Discussion (Line 496ff): “In summary, our data show a significant increase in transcripts encoding EPCRbinding and other A-type variants in parasites from severe and first-time infected patients, conversely transcripts of CD36-binding variants are found more frequently in parasites from non-severe and preexposed patients. Since CD36-binding variants are still overrepresented in all groups of adult malaria patients we postulate that the parasite population in first-time infected individuals may have broad binding potential after liver release as there is no pre-existing immunity to clear previously experienced PfEMP1 variants. During the blood stage infection selection towards EPCR-binding and other A-type variants, which may confer a parasite growth advantage and also increase the risk for severe malaria, may already have occurred in our adult severe malaria patients indicated by the longer period of infection.”

4. I don't think that additional experiments are needed to support the paper's claims. However, the clarity of the manuscript needs to be improved, so it can be understood by a broader readership.Introduction: The introduction is heavy on the var nomenclature. Although I appreciate that authors need to explain the var terminology, the introduction is very light on the specific research question. Moreover, they have not really placed the work in context to other attempts at var profiling. It is difficult for the reader to appreciate why it is difficult to profile var transcripts and why these genes are usually ignored in NGS approaches. This is one of the few studies that have attempted to assemble var reads from NGS. The reality is that NGS provides only a partial view of the var transcriptome. Most of the expressed var transcripts cannot be assembled into full-length contigs and contigs may represent different parts of the same gene. Here the authors have developed innovative bioinformatic approaches to classify the assembled var contigs (partial or otherwise) into different var groups, domains, and homology blocks by comparison to an annotated database of 2,400 sequenced parasite genomes and 235,000 annotated var genes. Also, there is almost nothing about the cohort in the introduction or why adult travelers are interesting to study. I think the manuscript will be much clearer if there is less var nomenclature in the introduction and more explanation of how the authors tackled this challenge of var profiling.

We are thankful for the detailed suggestions and have modified the manuscript accordingly:

Introduction: We included more information on adult severe malaria and our cohort as well as on the challenge of studying var gene expression in patient isolates:

– Line 58ff: “In particular, children under five years of age and pregnant women suffer from severe disease, but adults from areas of lower endemicity and non-immune travelers are also vulnerable to severe malaria. Both, children and adults are affected by cerebral malaria, but the prevalence of different features of severe malaria differs with increasing age. Anemia and convulsions are more frequent in children, jaundice indicative of hepatic dysfunction and oliguric renal failure are the dominant manifestations in adults (Dondorp et al., 2008; World Health Organization (WHO), 2014). Moreover, the mortality increases with age (Dondorp et al., 2008) and was previously determined as a risk factor for severe malaria and fatal outcome in non-immune patients, but the causing factors are largely unknown (Schwartz et al., 2001).”

– Line 114ff: “Due to the sequence diversity of var genes, studies of var expression in patients have relied on analysis of DBLα expressed sequence tags (EST) (Warimwe et al., 2009, 2012) informing on relative distribution of different var transcripts and qPCR primer sets covering some but not all subsets of DBL and CIDR domains (Lavstsen et al., 2012; Mkumbaye et al., 2017). So far, only very few studies (Tonkin-

Hill et al., 2018; Andrade et al., 2020; Duffy et al., 2016; Kamaliddin et al., 2019) have used the RNA-seq technology to quantify assembled var transcripts in vivo. Moreover, most studies have focused on the role of PfEMP1 in severe pediatric malaria. Consensus from these studies is, that severe malaria in children is associated with expression of PfEMP1 with EPCR-binding CIDRα1 domains (Jespersen et al., 2016; Kessler et al., 2017; Storm et al., 2019; Shabani et al., 2017; Mkumbaye et al., 2017; MagallónTejada et al., 2016), but elevated expression of dual EPCR and ICAM-1-binding PfEMP1 (Lennartz et al., 2017) and the group A associated DC5 and DC6 have also been associated with severe disease outcome (Magallón-Tejada et al., 2016; Avril et al., 2013, 2012; Claessens et al., 2012; Lavstsen et al., 2012; Duffy et al., 2019). Less effort has been put into understanding the role of PfEMP1 in relation to severe disease in adults, and its different symptomatology and higher fatality rate. Two gene expression studies from regions of unstable transmission in India showed elevated expression of EPCR-binding variants (DC8, DC13) and DC6 (Bernabeu et al., 2016; Subudhi et al., 2015), but also of transcripts encoding B- and Ctype PfEMP1 in severe cases (Subudhi et al., 2015).”

Discussion: Two additional paragraphs about our cohort and severe adult malaria (Line 388ff) as well as on *var* gene expression data from other adult malaria cohorts in relation to our data (Line 443ff) were integrated.

– Line 391ff: “Non-immune travelers and adults from areas of unstable malaria transmission are prone to severe malaria. Currently, scarce information on the PfEMP1-mediated pathogenicity responsible for the different symptomatology in comparison to pediatric severe malaria and the higher fatality rate in adults is available. Here we present the first in-depth gene expression analysis of 32 ex vivo blood samples from adult travelers using RNA-seq and expressed sequence tag analyses. Despite the relatively low number of patient samples recruited in 5 years, our data confirmed previously reported associations between transcripts encoding type A and B EPCR-binding PfEMP1 and infections in naïve hosts and disease severity (Table S6, S7)(Duffy et al., 2019; Tonkin-Hill et al., 2018; Kessler et al., 2017; Bernabeu et al., 2016; Jespersen et al., 2016; Lavstsen et al., 2012). Our results further suggests that parasite interaction with EPCR is linked to severe disease in children as well as in adults. However, since CIDRα1containing PfEMP1s possess multiple binding traits (Lennartz et al., 2017; Magallón-Tejada et al., 2016), co-interaction with other receptors may further increase the risk for severe malaria.”

– Line 446ff: “Data from adult cohorts are rather limited restricting our comparison mainly to three var gene expression studies based on qPCR or a custom cross-strain microarray (Argy et al., 2017; Bernabeu et al., 2016; Subudhi et al., 2015), but the majority of cases from the Indonesian RNA-seq study are also adults (Tonkin-Hill et al., 2018). A high expression of A- and B-type var genes and an association of DC4, 8 and 13 with disease severity has been reported for malaria cases imported to France (Argy et al., 2017) and parasites from severe Indian adults show an elevated expression of DC6 and 8 (Bernabeu et al., 2016) and DC13 (Subudhi et al., 2015). In the severe cases from Indonesia mainly the DCs 4, 8 and 11 were found on an elevated level (Tonkin-Hill et al., 2018). All studies are in agreement with the expression data from our cohort of adult travelers, although we here in addition found a higher expression of DC15 (EPCR binding) and DC16 (putative rosetting variants) var genes in malaria-naïve patients.“

5. Add Supplemental Overview Figure of methodology: Because of read depth, it is not possible to get the full-length sequence of most var genes and contigs may represent different parts of the same gene. A major technological advance here was leveraging the vast resource of sequenced and annotated var genes from thousands of parasite genomes to classify var gene fragments into domains and homology blocks and then make predictions about binding properties. It would be useful to provide a supplemental overview figure about how the two var profiling approaches (NGS and DBL amplicons) were fed into the system for bioinformatic classification.

We added two methodology figures. A general overview of both analysis approaches (NGS and EST=DBLa-tag) is now shown in Figure 2A. This will give the reader an overview about the concept of the different layers of *var* gene analysis used in this study. Furthermore, a more detailed Figure for the RNA-seq analysis including all the bioinformatic approaches used is presented in the Figure 2 —figure supplement 2 and we provide a Venn diagram of our patient cohort in Figure 1G.

6. The Figures and Tables relating to the var gene analysis are many and difficult to interpret without knowing the significance of the different domain types and homology blocks. Whereas analysis of domains and homology blocks are shown in Figures 4- 7, it is not until you come to summary Figure 8 that it shows how different domain types and homology blocks map to PfEMP1 subsets/adhesion types. Without this information, it is challenging to see how different var subsets differ before patient groups. The Figures needs to be more interpretable on their own without having to reference accompanying Tables. Consider moving the summary Figure 8 before Figures 4 to 7. Alternatively, introduce a var/PfEMP1 schematic in Figure 4. The schematic should introduce the concept of different var groups/PfEMP1 head structures/binding phenotypes/homology blocks. I recommend that you dispense with the terminology "segment level" from Tonkin-Hill 2018. There is already so much nomenclature for the reader to follow, plus many Figures and Tables to make sense of (9 Figures, 8 Tables, 10 Supplemental Tables).

To address this comment, we have moved Figure 8 (new Figure 3) showing a simplified version of the PfEMP1 scheme and a summary of the results from expression analysis to the beginning of the *var* expression analysis result section. The figure and the detailed legend will help the readers to understand the overall concept of *var* genes, their further subclassification and binding properties of the encoded PfEMP1 proteins. We have replaced the term “segment” by homology block in the entire manuscript (Materials and methods section ‘*var domain and homology block differential expression*’, Line 732ff).

7. Main Figures: In addition, it would help immensely if you would organize Figures 4-7, so that domains or homology blocks are arranged by predicted functional categories (e.g., DC8-EPCR, Group A-EPCR, Group A-nonEPCR, Group B or C-CD36). While this information is present in the accompanying Table, this information should be evident from the Figures, too. For instance, Figure 5 – Rearrange by var groups. Put the three DC8 domains together (DBLa2, CIDRa1.1, DBLb12), so it is easier to do a side-by-side comparison. For example, arrange by DC8-EPCR (DBLa2, CIDRa1.1, DBLb12), Group A-EPCR (e.g. DBLa1.7, CIDRa1.4, CIDRa1.7, etc), Group B or C-CD36. This will make it much clearer whether specific var subsets differ in the patient comparisons. Additionally, it would be more intuitive if you underline domains that target the same var group (DC8-EPCR, Group A-EPCR, Group A-nonEPCR, Group B or C-CD36) in panels B, D, E, and F. Having the information within the Figure will make it so much easier to follow. Figure 6 – Likewise, homology blocks should be arranged by var groups and add a label underneath the panels (e.g. block 584 = DC8, block 155 = NTSA). Figure 7 – Likewise, rearrange and label functional groups. To understand how you classified DBLa amplicons, I think that this figure also requires a schematic illustration at the top. As I understand it, you amplified a small tag from the DBLa domain and BLASTed against 235,000 annotated var genes to assign the tag to a PfEMP1/var group (DBLa0 = Group B or C, DBLa1 = group A, DBLa2 = DC8) and to predict the flanking NTS and CIDR domain types. Thus, this approach allows you to predict the PfEMP1 head structure from a small DBLa tag (not the full domain). In panel A, this approach was used to distinguish between Group A head structures that either bind EPCR (CIDRa1) or do not (CIDRb, y, or d). Furthermore, based on the DBLa amplicon read density, you are classifying the proportion of different var transcripts of different var groups/head structures. This approach would be more evident if you added a schematic at the top of Figure 7 showing the different types of PfEMP1 head structures, plus what was amplified and what was predicted by BLAST classification.

We have reorganized the data in all panels of the figures 4-7 as suggested (see also reviewer 1, minor comment #6). First, to make a better comparison between the different levels of *var* gene analysis, we have merged results according to the patient groups analyzed leading to the new Figures 4 (first-time infected versus pre-exposed) and 5 (severe versus non-severe). For all figures we have introduced a color code for (i) *var* gene groups (A: red, A-*var1*: dark red, A-*var*3: orange, B: blue, C: green, E-*var2csa*: yellow) and (ii) associated binding phenotypes (EPCR: beige, CD36: turquois, brown: unknown A-type, pink: ICAM-1, salmon: PECAM1, pale red: gC1qR). The color code enables the uniform labelling of transcripts, domains and homology blocks in the most correct way and accounts for the complexity of *var* genes/PfEMP1. We have also integrated the domain composition of the assembled contigs as well as the p-values into the Figures 4 and 5. Since all data from the tables 3-8 is now shown in the figures or can be easily found in the Supplementary Tables S9 and S10, we omitted the tables to avoid redundancy. Furthermore, we have grouped contigs of the conserved *var* variants *var1*, *var2csa* and *var3* and marked them with the color code scheme in the analysis on transcript level (Figure 4&5A,B). On domain level we labelled the associated *var* gene groups and binding phenotypes and arranged the domains in accordance to their DC. Moreover, a scheme showing the domain cassettes (DC) 1, 8, 13, 15, 16 was added showing how many domains of the DCs were found enriched in the different groups of parasite isolates. On homology block level we have indicated the associated *var* groups, binding phenotype and PfEMP1 domains either by using the color code and by extending the labelling.

Regarding the DBLa-tag approach: The limitation of this approach is, that it can only predicts the domain composition if a hit of the DBLa-tag is found in the database (requirements: at least 95% sequence identity over 200 bp). So, Varia can predict as much as the data allows, in our case about 85% of the DBLa-tags. This is now also explained in the methodology overview (Figure 2A) and therefore a scheme as suggested by the reviewer was skipped for this figure. But we also reorganized the figure in terms of introducing the color codes mentioned for the RNA-seq domain level analysis.

Minor comments:1. Other Table commentsTables 1 & 2 and cohort description:– More information is needed in the Methods and the Results sections about this patient cohort. In particular, the Methods should explain how patients were classified as severe and non-severe and whether WHO severity criteria were applied.

In line with the WHO criteria of severe malaria in adults we classified patients as having severe malaria if they showed clinical or laboratory signs of organ dysfunction or had extremely high parasitemia of over 10% or circulating schizonts upon hospitalization. Hyperparasitemia (>5%) per se was not sufficient for inclusion into the severe subgroup. We have clarified this in the Materials and methods section by inserting a whole paragraph on ‘patient classification according to severity’ (Line 618ff): “Severity was defined in line with the WHO criteria for severe malaria in adults (World Health Organization (WHO), 2014). Patients were considered as having severe malaria if they showed signs of impaired organ function (e.g., jaundice, renal failure, cerebral manifestations) or had extremely high parasitemia (>10%). In addition, patients #1 and #26 were included into the severe group due to circulating schizonts indicative of a very high sequestering parasite biomass associated with severity (Bernabeu et al., 2016) (Table S2).”

We refer to this paragraph in the Results section (Line 192ff): "Eight patients from the malaria-naïve group were considered as having severe malaria based on the predefined criteria. The remaining 24 cases were assigned to the non-severe malaria group (Figure 1G, Table S2)."

– In addition, the terms "cerebral dysfunction" and liver dysfunction" should be defined in the Methods. For instance, patient 9 in Table 1 has "liver dysfunction with highly increased transaminases with two upward arrows" but is labeled "non-severe". Is this symptom different from jaundice?

Thank you for pointing out this inaccuracy. We have reviewed the clinical information and altered the table (now Table S2) to include only criteria that qualified the patients for the classification of severe malaria. We have removed mention of the transaminases as these are not a criterium for severe malaria. The specific patient mentioned here had elevated transaminases but no jaundice or elevated bilirubin and was clinically relatively well. Therefore, he was not classified as having severe malaria.

– More information is needed in the Results section about the characteristics of this cohort. For instance, it is noteworthy that the cohort was all adult cases (Table 2) and all patients had febrile malaria (Table 1). Thus, they were all symptomatic. This information could go in the beginning of the Results (line 130 – Cohort characterization).

We totally agree and now described the cohort in more detail as suggested (Line 149f): “*This study is based on a cohort of 32 adult malaria patients hospitalized in Hamburg, Germany. All patients had fever indicative of symptomatic malaria.*”

Table 2– Explanation is needed for MSP-1 groups 1 to 4.

We changed the heading of the line in Table 1 into “Number of MSP1 genotypes [n (%)]” for explanation.

– Site of infection: Is it correct that one patient was infected in Germany?

Yes, actually this patient was infected via needlestick accident in a German hospital. To avoid confusion, we have now changed the term into “geographical origin of parasite isolates”. Since the naturally infected patient returned from Sub-Saharan Afrika, we have simplified our list underneath Table 1 stating now: “Geographic origin of the parasite isolates: Ghana (n=10), Nigeria (n=6), other Sub-Saharan African countries (n=15), unknown (n=1)”.

– Salmon RNAseq pipeline (lines 243-267, Figure 4 & Table 3)– This section is very confusing. As most var genes are not conserved across parasite genotypes it is important to explain to the reader that there are a few strain-transcendent variants in the parasite repertoire. Indeed, 8 of 15 contigs in the Salmon analysis belonged to strain-transcendent variants (var1, var2csa, var3). It would help to label the strain-transcendent variants in Figure 4.

To address this (please also refer to comment #6 from reviewer 1) we have now labelled conserved var variants in the Figures 4 and 5 and explicitly state in the section ‘Differential var transcript levels’ (Line 262ff): “We first looked for highly similar transcripts present in multiple samples. The Salmon RNA-seq quantification pipeline (Patro et al., 2017), which identifies equivalence classes allowing reads to contribute to the expression estimates of multiple transcripts, was used to estimate expression levels for each transcript. Due to the high diversity in var genes, mainly assembled transcripts of the strain-transcendent variants var1, var2csa and var3 were found to be differentially expressed.”

– Additionally, it would be good to say something about the 7 contigs that are not known as strain transcendent (e.g., NS_L1_88_Contig5431, Contig4767). What are these variants?

We already tried to address this in line 270ff of the result section: “Beyond the conserved variants, several var fragments from B- or C-type var genes were associated with a naïve immune status and three transcripts from A, DC8 and B-type var genes as well as var2csa were linked to severe malaria patients (Figure 4, 5, Table S9, S10).” Furthermore, the domain composition and associated var gene group and binding phenotype is now shown in the Figures 4 and 5 and can be found in the Supplement Tables S6. Results from BLAST of the contigs against the var database (varDB) and PacBio-assembled genomes can be found in addition to domain and homology block composition in Table S7.

– It is interesting that the two var1 alleles differed between patient subsets. Var1 has been considered a pseudogene with delayed transcription relative to other var genes, and less work has been done it.

Yes, we agree that this is a noteworthy observation which is already discussed in a whole paragraph (Line 477ff): “To the best of our knowledge, this study is the first description of expression differences between the two var1 variants, 3D7 and IT. The var1-IT variant was found enriched in parasites from first-time infected patients, whereas several transcripts of the var1-3D7 variant were increased in pre-exposed and non-severely ill patients. Expression of the var1 gene was previously observed to be elevated in malaria cases imported to France with an uncomplicated disease phenotype (Argy et al., 2017). In general, the var1 subfamily is ubiquitously transcribed (Winter et al., 2003; Duffy et al., 2006), atypically late in the cell cycle after transcription of var genes encoding the adhesion phenotype (Kyes et al., 2003; Duffy et al., 2002) and is annotated as a pseudogene in 3D7 due to a premature stop codon. Similarly, numerous isolates display frame-shift mutations often in exon 2 in the full gene sequences (Rask et al., 2010). However, none of these studies addressed differences in the two var1 variants that were recently identified by comparing var gene sequences from 714 *P. falciparum* genomes (Otto et al., 2019), and to date it is still unclear if both variants fulfill the same function or have the same characteristics previously described. Overall, the var1 gene – and the first 3.2 kb of the 3D7 variant in particular – seems to be under high evolutionary pressure (Otto et al., 2019) and both variants can be traced back before the split of P. reichenowi from P. praefalciparum and *P. falciparum* (Otto et al., 2018b). Our data indicate that the two variants, VAR13D7 and VAR1-IT, may have different roles during disease, however, this remains to be determined in future studies.”

– It also appears that mostly the 5' portion of var1 was transcribed (Table 1 and Figure S2). Do the authors have evidence if the full var1 transcript is being made or are these partial transcripts?

A common sequencing issue is the GC-bias leading to a higher coverage of GC-rich regions. We have already tried to address this by using the KAPA polymerase for amplification of the library, which is more or less dealing with that problem (Oyola *et al.*, 2012). Mapping of the RNA-seq reads to the two different *var1* variants shown in Figure 3 —figure supplement 1 indicates, that these transcripts are full-length, but with a better coverage at the 5’-end. Due the scaling the low coverage at the 3’end is difficult to display. Of note, for the *var2csa* gene coverage at the 3’-end was better than at the 5’-end.

– What is the difference between Contig2160 and Contig1987 in Table 3? Both include the NTSA-DBLa1.4 region of var1-3D7.

The contigs were de novo assembled from RNA-seq reads of different parasite isolates. The longer contig NS_L8_65_Contig2160 from patient isolate #29 and the shorter contig NS_L6_18_contig1987 from patient isolate #28. Both differ in length (7,253 bp and 548 bp) and pairwise sequence alignment reveals 7 mismatches (the sequences are provided in the Supplementary data S1). Since the *var1*-3D7 allele is particularly conserved in the 5’-region (Otto *et.al.*, 2019), this may also be caused by sequencing errors. The Salmon expression quantification algorithm allows reads to contribute to the expression estimates of multiple transcripts and is this able to account for this redundancy in the set of *var* transcripts.

– Why is Contig36393 assigned "var1" and not to the 3D7 or IT allele?

The contig contains only C-terminal domains of the *var1* gene, where both variants – 3D7 and IT – do not differ in domain composition. But pair-wise sequence alignment and BLAST reveal a substantial higher similarity to the *var1-IT* variant, so the contig was labelled accordingly in Figure 4

“NS_L1_80_Contig36393 γDBLγ8-DBLζ1-DBLε8 (*var1-IT*)”.

– Why does Table 4 have 5 contigs labeled "var1-3D7"? They appear to contain overlapping domains.

These contigs are assembled from separate samples. The Salmon expression quantification algorithm was used as it allows for redundancy in the reference database. Thus, multiple similar contigs originating from different samples can be called as differentially expressed. The results of the Salmon algorithm were compared with Corset, another algorithm that instead clusters similar transcripts and was found to give similar results. To make this clearer in the text we have added (Line 705ff): “As the RNA-seq reads from each sample were assembled independently it is possible for a highly similar transcript to be present multiple times in the combined set of transcripts from all samples. The Salmon algorithm identifies equivalence sets between transcripts allowing a single read to support the expression of multiple transcripts. As a result, Salmon accounts for the redundancy present in our whole set of var gene contigs from all separate sample-specific assemblies.”

2. Lines 145 and Lines 155-157: Specific patients are called out in the text "19-year old patient (#21)" and "patient #26", but these are not labeled in the figures.

This comment is in agreement with the minor comment 4 of reviewer 1. We have marked both patients in all panels of Figure 1.

3. Line 159 – This statement is confusing "the medical report showing that this patient from Jamaica was infected during his first trip to Africa". It is unclear when this patient was enrolled in your study. Are you saying patient #26 had prior malaria exposure on a previous African trip and was subsequently enrolled in this study from a new exposure to *P. falciparum* malaria?

We rephrased our sentence (Line 176ff): “The patient #26, positioned at the borderline to pre-exposed patients, was grouped into the naïve cluster in accordance with the Luminex data and the patient statement that this potentially pre-exposed patient returned from his first trip to Africa.”

4. MSP-1 genotyping was used to estimate the number of parasite genotypes. This should be mentioned in the Results section when referring to this genotype estimate.

We referred to this data shown in Table 1 by inserting the sentence (Line 150f): “MSP1 genotyping estimated a low number of different parasite genotypes present in the patients (Table 1).”

5. Line 167: Van Den Hoogen 2019 citation is missing in the references

Thanks for pointing this out. The reference is now correctly listed (Line 949ff).

6. Earlier work from your group and Lee et al. Nat Micro 2019 suggested that other non-var transcripts, such as KnobAssociated Histidine Rich Protein (KAHRP), were increased in severe malaria patients. Were any non-var transcripts involved in cytoadhesion and rigidity increased in severe malaria patients?

We already included expression data of genes involved in PfEMP1 biology (manually selected) in Supplementary Table S4 and S5 (“manual selection_PfEMP1 biology”), which is also referenced in the manuscript (Line 245ff): “In addition, we manually screened differentially expressed genes known to be involved in var gene regulation or correct display of PfEMP1 at the host cell surface (Table S4, S5).” In the first BioRxiv preprint version a detailed paragraph on this topic was included in the result and Discussion sections, but this was removed in order to streamline the manuscript. Nevertheless, we hope the interested reader might find our supplementary tables helpful. We didn’t observe a clear trend for KAHRP expression, even though we observed a slightly higher mean expression of KAHRP in the severe cases (logFC of 0.235), but this did not reach statistically significance. Instead, we observed significant higher expression of other PfEMP1 related genes, such as PTP1, PTP5 and SBP1.

Significance and Audience: This study is a technical tour de force and greatly extends the ability to investigate the *P. falciparum* var gene family in patient isolates, but it is not written in a manner that most readers will be able to interpret or appreciate the scientific advance because there is so much jargon. While the authors have done a heroic job of leveraging an enormous database of sequenced and annotated parasite genomes to classify assembled var contigs and DBLα amplicons into different functional groupings, the main take-aways will not be obvious to the general reader. I think this can be a highly influential study for its technical advance and because it more definitively implicates specific var subsets in malaria severity. As well, it addresses a relatively understudied adult travelers patient population. From a technical perspective, the tools and methods developed here will have great utility for future investigation of patient cohorts. There have been few attempts to profile var transcripts by NGS and the approaches developed here go much further beyond the author's previous work (Tonkin-Hill, Plos Biol, 2018). Put simply, it is extremely challenging to profile such a highly diverse gene family and this study offers a new and powerful approach to gain a more complete picture (still imperfect, but much improved).Conversely, some of the study conclusions are not strongly supported. For instance, there is no direct evidence that patients with first-time infection were selected "for parasites with more efficient sequestration" (Line 44) or "that the EPCR-binding phenotype confers more efficient sequestration of infected erythrocytes" (Line 127). Indeed, predicted CD36 var transcripts are highly expressed in all patients, so there is no evidence that the differences in circulating parasite ages are attributable to any given cytoadhesion trait. Moreover, alternative hypothesis for earlier sequestration have not been considered. Thus, a more balanced discussion is needed and the figures and texts should be improved to explain this study to a broader readership.Referees cross-commenting: Agree with the other two reviewers with the recommendation that no new experiments needed.Reviewer #3The investigators obtained malaria parasite samples from 32 adult travelers returning to Germany of which 10 had their first malaria episode, 9 had a history of malaria exposure while the malaria exposure status of 13 was unknown. They then generated anti-parasite antibody data using ELISA, luminex and protein microarray approach and then used the data to cluster the patients into malaria naïve and malaria exposed. They also used the clinical data to categorize the study participants into those with severe malaria and non-severe malaria. The parasite gene expression data was generated using whole genome transcriptome analysis approach and PCR amplification of var genes (DBLa-tag) and sequencing. The parasite gene expression data particularly var gene expression was related to host malaria pre-exposure and severity status. In summary, the study showed that parasites from naïve and severe cases express higher quantity of PfEMP1 containing the EPCR binding CIDRa1 domain compared to non-severe cases, while the pre-exposed and the non-severe cases expressed higher quantity of PfEMP1 containing domains known to mediate CD36 binding. In my opinion, the study methods and results have been described in details for expert in the field to follow but may be hard for non-experts because of the detailed analysis and data presented in the result section.The methods have been adequately described and the paper is generally well written but has lots of detailed analysis that can only be followed by experts in the field which probably not a problem. Below are my comments which I hope the authors will find useful.Major comments:1. The gap in knowledge the study is trying to fill has not been captured well in the introduction

To address this comment that was also raised by reviewer #2 we modified our introduction, trying to emphasize more clearly why it is difficult to profile *var* gene transcription especially in patient isolates and the advantages of our approach (see also comment 4 of reviewer #2).

2. Linking all PfEMP1 proteins containing CIDRa1 to EPCR-binding, in my opinion, is an over simplification of PfEMP1's cytoadherence versatility. There is no doubt that PfEMP1 containing CIDRa1 domain play an important role in disease pathogenesis and several field and in vitro studies referenced in the manuscript have confirmed this, but I think the evidence that all CIDRa1 bind EPCR is not strong. Again, no clear conserved EPCR binding motif has been identified in CIDRa1 domains unlike ICAM-1 binding DBLb for which a conserved motif has been identified. To be on the caution side, I suggest the name "CIDRa1 domain containing PfEMP1" be retained without linking them to EPCR binding function.

We disagree on this point. The molecular requirements for EPCR binding have been clearly shown by structural resolution, test of many CIDRa1 domains and unambiguous bioinformatics (Lau *et al.*, 2015; Rask *et al.*, 2010). Sequence identity-wise it is very easy to identify an EPCR-binding CIDR domain (even by eye!). CIDR domains have a unique compositions of sequence motifs (= homology blocks, HBs) (Rask *et al.*, 2010) and CIDRa1 domains with HB137 bind EPCR. The molecular basis of the few exceptions to this, the CIDRa1 variants not binding EPCR, are the *var1*-CIDRa1.2/3 and CIDRa1.5b. The molecular basis for these exceptions has been clarified and the reason is very clearly defined sequence wise. Apart from that, all parasites found to have HB137-containing CIDRa1 domains have bound EPCR. Only a single study found no EPCR binding of A types (Azasi *et al.*, 2018), whereas other studies with same parasites have demonstrated EPCR binding (Turner *et al.*, 2013; Lau *et al.*, 2015; Bernabeu *et al.*, 2016; Bernabeu *et al.*, 2019, Kessler *et al.*, 2017).

The mentioned ICAM1-binding DBLb motif is only specific for a subset (although the largest) of the group A type found DBLb domains. So not all A type ICAM1 binders have this motif and none of the group B type DBLb has this motif. Moreover, while EPCR-binding domains are rigid domains, DBLb domains change conformation in the ICAM1 binding, which may be related to the observation that no other trait of the DBLb domain can explain dome DBLb domains preference for ICAM1 or not.

3. I feel like there is no need to use the host immunity data to categorize the patients into malaria naive and preexposed. Instead, I suggest the antibody data, as a continuous variable, be correlated with the parasite gene expression to test the impact of host immunity on the phenotype (gene expression) of the infecting parasites. I think defining a binary variable (exposed and non-exposed group) using the host antibody data and then comparing their antibody response, as presented in Figure 1, is a circular analysis and not helpful.

Our rational for the categorization of patient samples into groups is based on the requirements for our differential gene expression analysis approach. The Luminex data were used for categorization of patients into first-time infected and pre-exposed, data from mADRB, ELISA and protein microarray validated our patient groups independently, but were mostly used to confirm patient #21 as first-time infected. The ultimate analysis, the random forest approach, was fed with data from all assays and clearly confirmed our subgrouping.

4. Severity of an infection is a factor of time (duration of infection), parasite virulence and host immunity. I think an analysis approach that allows testing of the relationship between the var expression (parasite virulence) and severity while adjusting for host immunity would have been better.

To address this and the previous point, we have additionally used the serological data as a continuous variable and plotted these PCR-reduced data from all assays against the expression of EPCR-binding CIDRa1 domains or CD36-binding CIDRa2-6 domains and calculated Spearman’s rank correlation coefficients (ρ). See Author response image 1. This analysis shows a weak negative correlation (ρ=-0.49) between host immunity and parasites expression of CIDRa1encoding transcripts; vice versa a weak positive correlation (ρ=0.36) was seen for host immunity and parasites expression of CD36 domains. However, this analysis has some limitations: (i) the severely ill patients were infected for a longer period and an already developed immune response during the acute infection may introduce a bias, (ii) a larger patient cohort would be necessary to see stronger correlations, but samples from travelers are rather rare (we sampled over 5 years only blood from 32 cases) and (iii) a higher expression of any PfEMP1 could also favor sequestration and severity, especially in adult cohorts with higher expression of CD36binding variants compared to children. To avoid confusions, we would suggest to leave this data point out of the manuscript.

Minor comments:1. What is the authors' interpretation of the fact that domains typical for CD36-binding PfEMP1 proteins are expressed at higher levels in malaria experienced while at the same time parasites from malaria experienced individuals also show elevated antibodies to the same PfEMP1 subset (Figure 9). Some few lines in the discussion will be helpful.

Since most PfEMP1 proteins have CIDRa2-6 domains capable to bind CD36, the parasite is able to express antigenically distant variants not recognized by the preformed immunity. We included a whole paragraph on CD36-binding in the discussion as also suggested by reviewer 2, major comment 2.

2. Any possibility of variation in var type expression over the asexual life cycle? And if yes, could the observed difference in parasite age between severe and non-severe cases be contributing to the observed variation in the expressed var subgroup between severe and non-severe?

Other studies have already shown that at least in NF54 parasites all *var* genes possess an almost identical expression pattern (parasite age of 5-25 hpi) (Dahlbäck *et al.*, 2007). A transcription pattern different from the other *var* genes is only known for the *var1* gene(s). In contrast to the expression seen in ring stages only for other *var* genes, *var1* is expressed throughout the different life cycle stages (see Discussion section, Line 482ff). This may be explained by a deletion of the intron responsible for repression upon parasite maturation in some parasite lines. We checked for a correlation between parasite age and *var1-3D7* expression in our sample set, but couldn’t find any. This is also clearly seen in the Figures, e.g., isolate #6 has a high abundance of early trophozoite stages (19 hpi) (Figure 2 —figure supplement 1), but we couldn’t detect any of the *var1* variants.

3. Line 478-482: the statement seems not to adequately explain why there are circulating relatively older parasites in non-severe cases since trophozoite stage parasites are expected to exist in both severe and non-severe cases. May be revise the statement to capture what you wanted to state.

We deleted the whole paragraph.

4. In my opinion, the word (direct ex vivo) in the title is not necessary.

We have modified the title to ‘*Gene expression profiling of adult malaria patients reveals common virulence gene expression in first-time infected patients and severe cases*’

5. Line 112: the sentence beginning with "No other domain has been…" needs to be revised.

The whole paragraph was modified and this particular sentence deleted.

Significance: This a descriptive study and the data generated confirms results of previously published studies and will be useful to Malariologist and those working in the field of malaria pathogenesis. I think results that confirm published findings, such as this study, are as important as those presenting novel findings. Overall, the conclusions are supported by the results and the methods used have been adequately described and presented. New experiments have not been recommended, instead an alternative analysis approach have been suggested (see comments).My expertise: I am a Malariologist with particular experience in the field of malaria pathogenesis.Referees cross-commenting: Went through the comments of the two other reviewers and I am happy to see that our observations are very close